

# Clouds over the summertime Sahara: An evaluation of Met Office Meteosat retrievals using airborne remote sensing

John C. Kealy[1†], Franco Marenco[1], John H. Marsham[2,3], Luis Garcia-Carreras[2], Pete N. Francis[1], Michael C. Cooke[1], and James Hocking[1]

[1]Met Office, Exeter, United Kingdom
[2]School of Earth and Environment, University of Leeds, Leeds, United Kingdom
[3]National Centre for Atmospheric Science
[†]Now at College of Engineering, Mathematics and Physical Sciences, University of Exeter, Exeter, United Kingdom

*Correspondence to:* John Kealy (J.Kealy@exeter.ac.uk)

**Abstract.**

Novel methods of cloud detection are applied to the unique Fennec aircraft dataset, to evaluate the Met Office derived products on cloud properties over the Sahara based on the Spinning Enhanced Visible and InfraRed Imager (SEVIRI) on-board Meteosat. Two cloud mask configurations are considered, as well as the retrievals of cloud-top height, and these products are

compared to airborne cloud remote sensing products acquired during the Fennec campaign in June 2011 and June 2012. Most detected clouds (67 % of the total) have a horizontal extent which is smaller than a SEVIRI pixel (3 km x 3 km). We show that, when partially cloud-contaminated pixels are included, a match between the SEVIRI and aircraft datasets is found in 80 ± 8 % of the pixels. Moreover, under clear skies the datasets are shown to agree for more than 90 % of the pixels. Cloud-top height retrievals however show large discrepancies over the region, which are ascribed to limiting factors such as the cloud horizontal

extent, the derived effective cloud amount, and the absorption by mineral dust.

## 1 Introduction

The challenge of observing atmospheric processes over the Sahara desert remains a fundamental obstacle in our understanding of the climate system of Africa, a key region for diagnosing global radiation budgets (Allan et al., 2007), aerosol transport (Schepanski et al., 2009), Atlantic hurricane activity (Dunion and Velden, 2004), and climate change (Giorgi, 2006). Much

of this difficulty arises from a lack of in situ observational networks across vast expanses of uninhabited, inhospitable desert, leading to difficulties in the investigation of important climatological features such as the Saharan heat low (Lavaysse et al., 2009). This observational deficit also has a large impact on the accuracy of numerical weather and climate models (Agustí-Panareda et al., 2010; Garcia-Carreras et al., 2013).

The Saharan Atmospheric Boundary Layer (SABL) is one of the deepest on Earth, often reaching 5–6 km by afternoon in

the summertime (Garcia-Carreras et al., 2015). Despite the extremely low relative humidities near the surface, the exceptional depth of the SABL can often lead to cloud formation at its top (Parker et al., 2005; Cuesta et al., 2009), with implications for radiative balance (Ramanathan et al., 1989) and dust transport (Birch et al., 2012). These clouds are typically the result



of thermal updrafts in the well-mixed SABL, and exhibit limited depth due to being capped by a strong inversion that marks the transition into the free troposphere (Cuesta et al., 2009). Observational case-studies show that albedo anomalies drive boundary-layer circulations in the Sahara and it has been hypothesised that these drive cloud formation (Marsham et al 2008; Cuesta et al 2009), but direct observations of this are limited (Ryder et al., 2015). Modelling studies by Huang et al. (2010)

and Birch et al. (2012) support this hypothesis, but the exact locations most favourable for cloud formation across the Sahara remain unclear.

Since ground-based observations in this sparsely populated part of the World are rare, satellite products are invaluable for investigating the Saharan atmosphere. The detection and transport of mineral dust in particular has been widely investigated using both geostationary satellites and polar-orbiters (Schepanski et al., 2007; Banks and Brindley, 2013; Banks et al., 2013).

Geostationary data, such as that provided by the Meteosat Second Generation (MSG) satellite, have proven very useful in these studies because of the high temporal resolution it offers for a given area, allowing the tracking of features as they cross the desert or pass into other geographical zones (Schepanski et al., 2012). Thus-far however, satellite-based studies specific to cloud over the Sahara have been limited to polar-orbiter data (Stein et al., 2011).

A variety of full-disk geostationary cloud products are readily available from the Satellite Applications Facility to support

Nowcasting and Very Short Range Forecasting (NWC-SAF, Derrien and Le Gléau, 2005; Derrien et al., 2013), the National Oceanic and Atmospheric Administration (NOAA, Schmit et al., 2002; Chang et al., 2010), the Met Office (Saunders et al., 2006; Hocking et al., 2011), and many other satellite applications facilities across the World. Recently these products have been the subject of cross-collaborations, under the framework of the International Clouds Working Group (ICWG). Before ICWG[1] was formed in 2014, a series of annual workshops known as the cloud retrieval evaluation workshops (CREWs) were

held, in order to discuss and intercompare the results of the products (Hamann et al., 2014; Roebeling et al., 2014). Such products include cloud mask, cloud-top height, cloud-top temperature, cloud optical thickness, and cloud effective radius, and the task is to validate them using observations from other sources; this is a necessary step to authenticate the robustness of the products, or reveal any weaknesses they may have (Derrien et al., 2005). This is particularly true in an environment such as the Sahara where a bright land surface and large atmospheric dust loads may make retrievals of cloud quantities challenging.

During CREW-4, the Sahara was identified as one of three areas in the SEVIRI full-disk image with the largest discrepancy between the members in an intercomparison of 12 different cloud mask products (Roebeling et al., 2014).

As part of the Fennec programme (Washington et al., 2012), the Facility for Airborne Atmospheric Measurements (FAAM) BAe-146 research aircraft was based in Fuerteventura (Canary Islands) in June 2011 and 2012 and performed 30 research flights (Fig. 1, Ryder et al., 2015). Fennec led to new quantification of the boundary-layer processes leading to clouds in the Sahara

(Garcia-Carreras et al., 2015; Ryder et al., 2015), demonstrated the role played by downdrafts generated from evaporation of precipitation falling from clouds over the Sahara (Marsham et al., 2013), and showed that the day-to-day variability in top-of-atmosphere radiative heating is largely a function of total column water vapour and not dust, suggesting that clouds may be key to the radiative balance of the Saharan Heat Low (SHL, Marsham et al., 2016).

---

[1]http://www.icare.univ-lille1.fr/crew/



The aim of the present paper is to analyse and validate the Met Office cloud products based on Meteosat over the Sahara, by comparing the satellite dataset to airborne cloud retrievals. A description of both the airborne and satellite datasets is provided in Sect. 2, along with a description of the methods used. Two Met Office cloud products for Meteosat are then investigated individually, the cloud mask product (Sect. 3.2) and the cloud-top height product (Sect. 3.3). We also consider the ideas

proposed by Marsham et al. (2008) and Cuesta et al. (2009) regarding the role of surface albedo and topography in cloud formation, within the context of these datasets (Sect. 3.4). Sect. 4 discusses the results and Sect. 5 summarises our conclusions.

## 2   Datasets

### 2.1   Satellite derived products

The cloud comparison described in this paper is based on data from the Meteosat Second Generation (MSG) Spinning Enhanced

Visible and InfraRed Imager (SEVIRI) and derived cloud products produced by the Met Office operational systems (Saunders et al., 2006). SEVIRI generates an image every 15 minutes using twelve spectral channels (Schmetz et al., 2002) and within the latitude band of interest in this study (approximately 15° N – 25° N) the spatial resolution is close to 3 x 3 km$^2$.

In order to derive information from SEVIRI, the first step is to identify which pixels contain cloud. The cloud mask developed at the Met Office by Hocking et al. (2011) is chosen here as the basis for comparison with the aircraft cloud data. This mask

is derived by applying a variety of threshold tests which use the observed brightness temperatures ($T_B$), reflectances, and simulated clear-sky radiances to infer whether cloud is present in each SEVIRI pixel. These tests are summarised in Table II of Hocking et al. (2011) along with a description of the scheme and a validation against both the Derrien and Le Gléau (2005) cloud mask for SEVIRI and the MODIS cloud mask.

Here we use two separate configurations of the cloud mask. The first detects a cloud for a given pixel if one or more of

any of the tests in Hocking et al. (2011) returns "true". It is intended primarily for applications that require a cloud-free pixel, where even the effects of a small amount of cloud-contamination is undesirable, and we therefore refer to this configuration hereafter as the "*AllCloud*" mask. The second configuration of the cloud mask, hereafter referred to as the "*CloudRetrieval*" mask, is intended as a detection tool for those pixels where it is feasible to carry out cloud property retrievals. Pixels which correspond to low fractional coverage or low cloud optical thickness are less radiatively significant in the infrared, and an

attempt to retrieve a cloud parameter (e.g. cloud-top height or cloud-top temperature) for these pixels can produce misleading results. A subset of the tests described by Hocking et al. (2011) have a general tendency to identify such pixels. These consist of three types of spatial coherence test using the 0.8 μm, 10.8 μm and HRV channels, and a test known as the 15-minute temporal differencing test. The spatial coherence tests work on the principle that cloud edges tend to exhibit large standard deviations in their brightness temperature and reflectance values, while the 15-minute temporal differencing test identifies a pixel as cloudy

when its brightness temperature decreases by more than a predetermined threshold value since the previous observation time. The radiatively less-significant cloudy pixels (i.e. those which are flagged only by one or more of these four tests and not by any other cloud test) are excluded from the *CloudRetrieval* mask (whereas they have been included in the *AllCloud* mask along





with all other cloud-flagged pixels, as noted above). For this reason the *AllCloud* mask will always contain more cloudy pixels than the *CloudRetrieval* mask.

The second product investigated here is cloud-top height (CTH). In order to calculate satellite-derived CTH, the algorithm incorporates numerical weather prediction (NWP) model profiles and performs radiative transfer calculations, which in this
study are based on the Met Office Unified Model (MetUM). Over the African continent, the Global Model (GM) is currently the chosen configuration of the MetUM used in the cloud retrievals. By inputing vertical atmospheric profiles from the GM into a fast radiative transfer model, radiance information can be simulated at discrete levels throughout the depth of the atmosphere. The radiative transfer model used at the Met Office is known as RTTOV (Saunders et al., 1999).

In determining the cloud-top height (CTH), the Met Office algorithm uses three spectral channels in the infrared, 10.8 µm,
12.0 µm and 13.4 µm. Firstly, a slightly updated version of the minimum residual technique described in Eyre and Menzel (1989) is employed. This method uses a two-parameter state vector, the active variables being the cloud-top pressure $p$ and the effective cloud amount ($N$), with the latter being the product of the true cloud fraction and the cloud emissivity (and which is assumed independent of wavelength). For each RTTOV model level, we use an error-weighted version of Eyre and Menzel's Eq. (4) to derive a profile of best-fit effective cloud amount for each cloud-top pressure $p$:

$$N = \frac{\sum_j (R_j^m - R_j^c)[R_j^o(p) - R_j^c]/\sigma_j^2}{\sum_j [R_j^o(p) - R_j^c]^2/\sigma_j^2} \tag{1}$$

where $R_j^m$ is the measured radiance in channel $j$, $R_j^c$ is the RTTOV calculated clear sky radiance in channel $j$, $R_j^o(p)$ is the RTTOV calculated blackbody radiance at pressure level $p$ for channel $j$, and $\sigma_j$ is the variance for channel $j$. Using these values of $N$, a profile of minimum residual cost ($J_{mr}$) is constructed for each $p$ value:

$$J_{mr} = \sum_j (R_j^m - [R_j^c(1 - N) + N R_j^o(p)])^2/\sigma_j^2 \tag{2}$$

A stability cost is then added to the minimum residual cost based on the temperature lapse rate calculated using the MetUM, and the value of $p$ corresponding to the minimum of the resulting cost profile is selected as the cloud-top pressure. Generally the minimum residual scheme is intended to find suitable solutions for mid- to high-level clouds.

If a solution is not by found by the minimum residual scheme (for example, if the resulting solution variance is greater than a pre-determined threshold, or if the resulting cloud-top is put at a relatively low height and hence considered to be unreliable),
a method known as the "stable layers scheme" is attempted, the second of three component schemes within the algorithm. This scheme employs only the 10.8 µm channel, and attempts to match the observed radiance from SEVIRI ($R_{10.8}^m$) with each vertical level in RTTOV ($R_j^o(p)$). Similar to the minimum residual scheme, an additional cost is applied to reduce the likelihood of solutions placing the cloud-top at the bottom of an unstable layer, since this could potentially cause unrealistic convection if the output were to be subsequently assimilated into an NWP model. This background instability cost is given a
much higher weighting for the stable layers retrieval compared with the previous minimum residual retrieval.





If the stable layers scheme also fails (for example, if a suitable cloud-top which satisfies the stability constraint and is also consistent with the measured radiance cannot be found, or if the resulting cloud-top is put at a relatively high altitude and hence considered to be unreliable), then a final method is applied which makes a direct comparison between the observed ($R_{10.8}^m$) and the simulated value, assuming an opaque cloud layer, and with no account taken of atmospheric stability or cloud transparency.

This is known as the "profile matching scheme", and is always applied if both the stable layer and minimum residual schemes fail to find a solution (which is typically about 4 % of all SEVIRI pixels in the full disk). A value for the effective cloud amount ($N$) has been assigned to each pixel within the profile matching scheme (this value is derived from the minimum residual scheme). Since cloud is assumed to be opaque in this scheme, this $N$ value does not have the same quantitative meaning as the equivalent values found within the minimum residual scheme. However, it is nonetheless retained here for its qualitative value,

because useful information about cloud properties within a pixel can still be inferred from the $N$ value originally assigned to it.

### 2.2    Airborne cloud dataset (cloud under aircraft)

During the Fennec campaign, 24 out of the 30 FAAM BAe146 aircraft flights were carried out into the remote Sahara of Mauritania and the western part of Mali (Fig. 1 and Table 1), while the other six flew mainly over the ocean. Much of the

flying time was spent at cruising altitude (~8,000 m above sea level) when the aircraft was transiting from the operating airport to the target area. During this time, vertical profiles of the atmosphere below the aircraft were sampled with the on-board lidar, and measurements of upwelling radiance were obtained with the Heimann radiometer. These two datasets are principally investigated here, as they provide remotely-sensed cloud information. The meteorological conditions during Fennec, details of each individual flight's location, and principal objectives are described in Ryder et al. (2015).

A dataset of aircraft cloud mask and CTH retrievals for each FAAM flight using the Leosphere ALS450 backscatter lidar and Heimann KT19 radiometer has been derived. The lidar has an operational wavelength of 355 nm with daytime capability, and a nadir viewing geometry (Marenco et al., 2011; Chazette et al., 2012). The full overlap between the emitted light and the receiver field of view (overlap range) is estimated at 300 m. Lidar signals were acquired with a vertical resolution of 1.5 m and an integration of 2 s; to improve the signal-to-noise ratio, the signals have been further smoothed to a vertical resolution

of 45 m. The lidar qualitative plots show in general a significant dust load in the first ~6,000 m of the atmosphere from the surface, consistent with Cuesta et al. (2009), although no attempt has been made to invert the data to extinction coefficient, as this requires a time consuming, non-automated procedure.

Configured in the infra-red spectral range of 8 to 14 μm, the Heimann radiometer measures upwelling radiation at a temporal frequency of 1 Hz, providing a high-resolution dataset of brightness temperature across a 22° field of view around nadir. The

dataset of the Heimann radiometer has been integrated to yield a 2 s resolution, time-matched with the lidar dataset, thus allowing the two datasets to be considered together. At typical aircraft speeds, this integration time is equivalent to a horizontal resolution of approximately 300 m (along-track).

A cloud detection and cloud-top retrieval algorithm has been applied to each lidar vertical profile, making use of the uncalibrated range-corrected signal profile P(R), using the thresholds given in Allen et al. (2014). The basis for this algorithm lies in



the detection of intense peaks with large gradients in the backscattered range-corrected signal. A cloud is identified if (a) P(R) > 4000; (b) P(R) > 3 × P(R - 200 m); and (c) no other cloud-top is found in the same profile between R - 500 m and R, where R is the range below the aircraft. Once a cloud is detected, the cloud-top range $R_c$ is determined as the first lidar measurement point, starting at R and moving inwards to (R - 200 m), where P($R_c$) < 1.5 P(R - 200 m). The cloud-top range $R_c$ is then converted to

CTH. As an additional quality control, cloud-top heights under 4500 m have been discarded, because the lidar signal tended to be noisy at low altitude due to attenuation by dust, and the noisy signals have shown to lead to a number of false detections. The assumption that no clouds occur under 4500 m is reasonable over the Sahara, where the summer atmosphere is hot and dry in daytime and clouds are found at the top of a very deep boundary layer (Stein et al., 2011; Marsham et al., 2013). In all locations where a CTH has been determined, a lidar cloud flag has been set, indicating detection. Note that this algorithm

detects the cloud-top spike, and would lead to a missed detection when the aircraft is flying in the cloud or when the cloud-top is within the overlap range; additional rare missed detections were encountered where the cloud cover was partial within the integration time used.

To address the missed detections, a second cloud detection algorithm has been applied using the Heimann radiometer brightness temperature ($T_B$) dataset. First of all, a clear-sky baseline $T_B$ was derived, using the lidar cloud flag to select cloud-free

data, and smoothing to remove the effect of outliers. The baseline $T_B$ is determined as follows: (1) pixels flagged as cloudy by the lidar and pixels measuring a $T_B$ < 290 K are discarded; (2) data points have been grouped in 10 minute intervals; (3) points outside 1 standard deviation of the local group mean have been considered outliers and also discarded; (4) the remaining data in each group have been averaged; and (5) the cloud-free $T_B$ calculated in this way for each group has been interpolated back to the dataset resolution by fitting a cubic spline with a 4-point neighbourhood. Cloud detection has then been based on

the difference between the baseline $T_B$ and the measured $T_B$. A difference greater than 3 K has been considered an indication of cloud presence, and has been used to set a Heimann cloud flag. Note that this cloud flag, being sensitive to $T_B$, permits to resolve most of the missed detections by the lidar, including those when flying in cloud or very near its top. As an additional quality control, clouds detected when flying lower than 4500 m have been omitted for the same reason that they were omitted from the lidar dataset. Data acquired during aircraft turns involving a bank angle larger than 10° have also been omitted, as the

Heimann $T_B$ would be tilted from the vertical and affected by the different viewing geometry.

The two cloud flags are then combined into an aircraft cloud mask, with a cloud being flagged as detected if either the lidar or the Heimann indicates that a cloud is present. The aircraft cloud mask has been reviewed by looking at the lidar qualitative plots and the Heimann $T_B$ plots, and the thresholds have been adjusted until we have been satisfied that we have a robust product. Note however that the Heimann method may well be dependent on the season and the geographical region (the Sahara

being hot with an extremely deep boundary layer in the summer enabling a significant thermal contrast between the clouds and the underlying surface); therefore we do not expect it to work as well outside the specific Fennec dataset.

At the end of this procedure, we end up with the following data on a ~300 m horizontal resolution (along-track), based on the airborne data:

(a) aircraft cloud mask (also articulated into lidar cloud flag and Heimann cloud flag): indicates the presence of a cloud under

the aircraft;





(b) lidar CTH, only for those points that indicate cloud detection from the lidar;

### 2.3  Airborne cloud dataset (cloud above aircraft)

Downwelling short wave irradiance measured by a modified Eppley pyranometer (a Broad-Band Radiometer, BBR, calibrated for the spectral range 0.3 μm - 2.8 μm) is used to detect clouds above the aircraft. Using a ground-based approach, it has been
shown that pyranometer data is independently capable of differentiating clear-sky conditions from cloud (Long and Ackerman, 2000; Xia et al., 2007). The BBR produces a noisy field with rapid, large-amplitude variations in signal when the incident solar radiation passes through cloud. By differentiating the radiometer's time series, and thresholding at 7 W m$^{-2}$ s$^{-1}$, a cloud filter was generated. This threshold was fine-tuned through a trial-error-process. Neighbouring points collected within approximately one minute (~10 km) of those flagged by the BBR method were also flagged, to fill inherent gaps in the filter. Since data-points
for which cloud lay between the aircraft and the satellite field of view are not useful for a quantitative comparison with MSG, this filter was used to remove these points from both the SEVIRI and aircraft datasets entirely (Table 1).

As with the Heimann cloud mask, the BBR-based cloud detection algorithm is sensitive to roll and pitch angles. Therefore, aircraft points which deviated from a nadir-zenith alignment by more than 10° have been removed from the comparison.

### 2.4  Additional data

Throughout the 30 BAe146 flights of Fennec, a total of 121 Vaisala RD94 dropsondes were deployed from the aircraft. These are used here as supplementary information on atmospheric thermodynamic profile for specific times during the flights. Also used are images taken from the standard rearward facing digital camera of the BAe146. Topography and surface albedo maps were provided using data from the GLOBE digital elevation model (Payne et al., 1999) and MODIS satellite data (Gao et al., 2005) respectively.

### 2.5  Spatial and temporal matching of datasets

To allow for a direct comparison of the satellite and airborne datasets, a mean value was derived for both the aircraft cloud mask and aircraft CTH within each SEVIRI pixel (the number of aircraft points falling within each satellite pixel varied between 2 and 17, with an average of ~8). This mean value of aircraft cloud mask within each SEVIRI pixel is expressed as an aircraft-derived estimate of cloud fraction, with the value of this "aircraft cloud fraction" represented by a number between 0 and
25  1.

Since the acquisition time for each SEVIRI frame is 15 min, we evaluate the inherent measurement uncertainty arising from a changing cloud field within this time-step. The actual time difference is assumed to be half of the 15-min uncertainty, because each aircraft point has been matched with the satellite frame nearest to its time-stamp (within either 7½ min before or 7½ min after the middle of SEVIRI's full-disk scan). We have translated the uncertainty from temporal to spatial coordinates by cal-
culating the percentage of pixels in the cloud mask which change from cloudy to non-cloudy or vice versa during a timestep. A single value of uncertainty was then derived for each flight by averaging these percentages across each pair of successive



frames. The domains used in this method were chosen by defining a rectangle bounded by the maximum and minimum latitude/longitude points along the BAe146 flightpath for each individual flight, and using acquisition times in accordance with the flight's duration. This was done to aid consistency in the comparison, however, we emphasise that the uncertainty itself was derived from SEVIRI data only.

5 Changes between subsequent cloud mask frames tend mainly to occur near to cloud boundaries, as one would expect, and so it is noted that scenes with more broken/cellular cloud structures will inherently carry a greater temporal matching uncertainty. These uncertainty values are included in the analysis for each flight (Tables 2 and 3). In addition, a combined overall uncertainty across all the flights has been derived. This overall value for the entire dataset excludes flights for which the cloud coverage was less than 10 %, since the uncertainty value can be biased by cloud-free (or near cloud-free) conditions.

## 3 Results

### 3.1 Cloud scale analysis

The aircraft along-track resolution of approximately 300 m permits us to estimate the horizontal extent (calculated by sampling the number of adjacent pixels along the aircraft track) of clouds in the direction of travel; results of this analysis are shown in Fig. 2. A large fraction of the clouds encountered by the aircraft (67 %) are actually smaller in scale than the 3 km-wide SEVIRI pixels, and are therefore filling these pixels partially. In fact, Fig. 2 shows that the smallest cloud size identifiable by the BAe146 (300 m) is also the size most frequently observed (25 %), implying that one quarter of the entire dataset is made up of clouds with a horizontal extent less than (or equal to) the aircraft resolution. Therefore, radiation reaching SEVIRI will often have a contribution from both the altitude of the cloud-top and from the desert surface below, all within the same pixel. This can have varying implications, which will be investigated throughout the following sections.

### 3.2 Cloud mask comparison

A direct time-series comparison for flights B602, B608, B614 and B706 is presented in Fig. 3. These flights have been chosen as examples of scenes that have sufficient cloud amounts to make a qualitative comparison. In general, the visual comparison of cloudy SEVIRI pixels against the aircraft cloud mask is encouraging. For flight B602 (Fig. 3a), 16 % of the SEVIRI pixels are cloudy in both configurations of the cloud mask, compared to 14 % cloudiness in the aircraft points. Similarly, both B608 and B614 (Fig. 3b and 3c), show a difference of less than 3 % between the aircraft and *AllCloud* masks. The *CloudRetrieval* mask exhibits a lower percentage of cloud coverage compared to the aircraft for these two flights, but the difference here is still no more than 6 %.

In the case of flight B706 (Fig. 3d), a cloudier scene appears from the aircraft dataset than with SEVIRI, with nearly twice as much cloud detected (26 %) compared with the *CloudRetrieval* mask (14 %). The digital camera footage (Fig. 4a) shows that the cloud during this flight was generally stratiform in its structure, but also exhibited numerous gaps which allowed a contribution of upwelling radiation from the desert floor. Overall however, such an underestimation of the cloud field in





SEVIRI was only observed in a small portion of flights. The full comparison for each flight using this method can be found in Table 1.

Figure 5 shows how a pixel by pixel comparison of the SEVIRI cloud mask with the aircraft cloud mask is distributed as a function of aircraft cloud fraction for flights B608 and B706. These particular flights have been selected for individual

analysis because of the consistent altitude of cloud-tops apparent in the aircraft data, the absence of cloud detected above the aircraft, and for B608, the larger dataset (due to the BAe146 remaining at cruising altitude throughout the whole flight). Since *AllCloud* allows for partially cloud contaminated pixels, the ideal *AllCloud* histogram of Fig. 5 would show 100 % of the cloud-free pixels corresponding to an aircraft cloud fraction less than 0.1, and all of the cloudy pixels spread about the aircraft cloud fraction range 0.1–1.0. Conversely, an ideal version of Fig. 5 for *CloudRetrieval* would show 100 % of the cloudy pixels

matching with an aircraft cloud fraction greater than 0.9.

The cloud structure of B608 is illustrated by the digital camera image at 1616 UTC (Fig. 4b). As Sect. 3.1 highlights, the broken, small scale nature of these clouds means that care must be taken when interpreting a comparison with SEVIRI cloud flags, each one of which represents an area of ~3 x 3 $km^2$. In the case of the *AllCloud* mask for B608 (Fig. 5a), 87 % of clear-sky SEVIRI pixels have been matched with an aircraft cloud fraction of less than 0.1. Since *AllCloud* has been designed

with clear-sky pixels in mind, this result implies a good agreement between the aircraft and MSG datasets. In terms of cloud, 63 % of cloudy SEVIRI pixels have been matched with an aircraft cloud fraction exceeding 0.9, to which can be added the 19 % of cloudy flags matched within the aircraft cloud mask range 0.1–0.9. The remaining 18 % of cloudy pixels along the B608 flight path have been matched with an aircraft cloud fraction of less than 0.1.

It is not expected that SEVIRI should flag cloud with an aircraft cloud fraction below 0.1. Part of this discrepancy can

be attributed to the temporal uncertainty induced by a changing cloud field during SEVIRI's scan time (calculated as 16 % for B608, see Sect. 2.5), it is also likely that the small horizontal extent of the clouds in this scene may have an effect on SEVIRI's cloud detection scheme. In addition, the one dimensional nature of the aircraft flight path means that although the datasets should agree for clear sky pixels, the aircraft may miss cloud observed by SEVIRI which lies parallel to, but not along, its track. The results of this method of matching SEVIRI *AllCloud* cloud mask flags with aircraft cloud fractions for each

individual BAe146 flight is shown in Table 2.

The *CloudRetrieval* cloud mask is more conservative in its approach to flagging cloud, and partly as a result of this, Fig. 5b shows a higher percentage of cloudy SEVIRI pixels which are matched to aircraft cloud fractions exceeding 0.9. However, unlike *AllCloud*, aircraft cloud fractions in the range 0.1 – 0.9 are not intended to be matched with a cloudy flag in the SEVIRI data. This means that for B608, the number of SEVIRI pixels flagged as cloudy by *CloudRetrieval* match with 75 % of the

aircraft dataset, in contrast to 82 % from *AllCloud*. The number of clear-sky flagged pixels for *CloudRetrieval* that matched with an aircraft cloud fraction below 0.9 was 96 %.

B706 observed a more uniform cloud layer than B608 during the flight (Fig. 4a), which mostly lay in the region of the Mali-Mauritania border. 85 % of SEVIRI cloud flags detected by *AllCloud* were matched with aircraft cloud fractions above 0.9, with this figure rising to 91 % for *CloudRetrieval*. As was implied in the qualitative analysis shown in Fig. 3d, SEVIRI

tends to detect less cloud than the aircraft, with 9 % of the *AllCloud* and 12 % of the *CloudRetrieval* cloud-free flags showing





aircraft cloud fractions above 0.9. Nonetheless, these values are within the temporal uncertainty (15 % for *AllCloud* and 9 % for *CloudRetrieval*), and we therefore consider the datasets for B706 to be in excellent agreement.

Across all 24 flights and using every valid data-point, we find that 91 % ± 8 % of pixels where SEVIRI did not detect a cloud in the *AllCloud* mask matched with an aircraft cloud fraction of less than 0.1, showing excellent agreement between

the datasets for clear skies (Fig. 6a). For the pixels flagged as cloudy, 59 % of pixels with a SEVIRI *AllCloud* cloud detected corresponded with an aircraft cloud fraction greater than 0.9. Combining this with all other aircraft cloud fractions which were greater than 0.1, we find an overall value of 80 % ± 8 % for the *AllCloud* mask. As for the *CloudRetrieval* configuration of the cloud mask (Table 3), 68 % ± 5 % of cloudy SEVIRI pixels matched with aircraft cloud fractions of greater than 0.9. Since the *CloudRetrieval* mask is not expected to account for cloud fractions of 0.1–0.9, this seems to flag a 32 % overestimation of cloud

pixels. As for the identification of clear-skies by *CloudRetrieval,* 97 % ± 5 % of all pixels flagged to be non-cloudy are matched with an aircraft cloud fraction below 0.9, again implying very good agreement (since non-cloudy pixels in *CloudRetrieval* include partially the cloudy pixels).

Overall, these results suggest that although the datasets match well in general, the SEVIRI detection scheme seems to overestimate cloudiness, as evidenced by the 20 % of cloudy pixels from *AllCloud* with aircraft cloud fractions of less than

0.1, and the 32 % of cloudy pixels from *CloudRetrieval* with aircraft cloud fractions less than 0.9.

### 3.3   Comparison of cloud-top height

Fig. 7a illustrates how the aircraft and MSG cloud-top heights correlate. The broad range of cloud-top height values retrieved by SEVIRI, spanning from below 500 m to above 14,000 m, is in sharp contrast with the narrow range in lidar retrieved CTHs, which is limited to 6,000–8,000 m, with the lidar heights consistent with the typical summertime SABL depth. Note that cloud

lying above the aircraft flight path has been filtered out using the BBR, and the dryness of the lower SABL makes the lower clouds very unlikely. The extremes are therefore assumed to be incorrect retrievals from the same clouds observed by the lidar at the top of the SABL. The result shows therefore that over the Sahara, CTH retrievals from MSG may differ substantially from reality.

To understand the CTH errors, we split the MSG retrievals into the component schemes from which they are calculated (Fig.

7b). From this figure, a strong contrast is apparent between the schemes used above and below 6,000 m. The minimum residual scheme dominates for clouds above 6,000 m, as would be expected, while the profile matching scheme dominates below 6,000 m. The stable layers scheme is used only a small fraction of the time, however, we note that all of the points which do use this scheme lie at an altitude broadly consistent with the aircraft data (~6,000 m).

From Fig. 7b, it is apparent that in this dataset the profile matching scheme returns CTHs between 1,000 m and 6,000 m.

Given the extremely low relative humidity within the SABL, particularly near the surface, low CTH's are however not expected in this region. In addition, clouds are detected in the same location but at a higher altitude with the lidar. To illustrate these considerations, Figure 7c shows dropsonde and MetUM profiles from flight B608 at 1611 UTC, a time when some of these low CTHs from SEVIRI appear; the profile exhibits the characteristic deep and well-mixed dry adiabatic profile within the boundary layer depth. The good agreement of this model profile with the dropsonde gives us confidence that the meteorological input



to the RTTOV radiative transfer model is not the primary factor driving the profile matching scheme towards unrealistic CTH results. Both the dropsonde and model profiles suggest an approximate CTH range of ~5500–6500 m (450-500 hPa), in line with lidar observations (nearest lidar derived CTH to the dropsonde location was 5927 m). In contrast, the closest SEVIRI retrieval to this location returns a CTH value of 1820 m (with retrieved effective cloud amount $N$=0.5). It should be noted that

more often than not, the NWP profiles do not match so well with reality. However as this particular case shows, CTH errors of this magnitude can still arise even when the NWP profiles are a close match to the true atmospheric state.

In general, the profile matching scheme accounts for ~4 % of all the pixels in the SEVIRI full-disk. However, over the Sahara we find that this number rises for many of the flight days, in one case exceeding 20 % of the domain shown in Fig. 1 (during flight B608). For the profile matching scheme, which is shown here to frequently underestimate CTH, it is likely that

the assumption of opacity in cloud is not valid for many of the pixels. If contributions from below the cloud base are affecting the retrieval, then the matched radiance flux may be overestimated, and will hence retrieve a warmer (and therefore lower) cloud-top. This can be linked to a more general issue in cloud retrieval over the Sahara; a large portion of the cloud (67 % in this study) has a horizontal extent smaller than the SEVIRI resolution. Therefore a contribution from the desert surface within each cloudy pixel will be commonplace, and the expected effect of introducing contributions of the warm desert surface would

indeed be an observed decrease in altitude of the retrieved cloud-top.

Points in Fig. 7a have been coloured by the retrieved effective cloud amount ($N$) value for each SEVIRI pixel. The majority of clouds appearing above 10,000 m have retrieved $N$ values below 0.5 (cyan and yellow). This is also evident in Fig. 7b. Likewise, all clouds with SEVIRI retrieved tops below an altitude of 3,000 m have an $N$ value of less than 0.5. In the case of the low clouds retrieved using the profile matching scheme, the $N$ value should be used with caution, because this value is not

explicitly calculated (due to the assumption of cloud opacity in this scheme), and is instead taken from the minimum residual scheme's algorithm. Nonetheless, Fig. 7b implies that as the value of $N$ increases, the SEVIRI-retrieved CTH also increases, tending towards 6,000 m (the approximate top of the SABL) as the $N$ value tends to 1. Similarly for the higher clouds, Fig. 7b implies that as $N$ increases for points using the minimum residual scheme, the SEVIRI-retrieved CTH tends to decrease.

As a specific example of this effect, Fig. 7d shows the CTH retrievals for flight B608. Points with an $N$ value above 0.7 are

highlighted in red and green. These points show a good fit with the lidar data (mean difference lidar CTH – SEVIRI CTH for B608 is 606 m), bearing in mind that the number of pixels used with $N$ values above 0.7 is limited (45 pixels). For pixels with $N$ values below 0.7, a large spread in cloud-top height (mean difference for B608 is 4402 m) is present in SEVIRI, with the highest retrieved cloud exceeding 13,500 m (corresponding to a nearby lidar CTH of 6650 m).

### 3.4 Cloud distribution over the Sahara

Despite a mild tendency to over-flag cloud at times, the results of this study suggest that the Met Office cloud mask is relatively robust. A potential application could be its use for a cloud climatology of the Sahara desert from SEVIRI data. Because MSG lies in a geostationary orbit, such a climatology could expand on previous studies (Stein et al., 2011) to include cloud evolution, diurnal variation, and the location of areas most prone to cloud development.



A first look at the most typical locations of cloud over the western Sahara has been compiled from the SEVIRI data used in this study (Fig. 8a). This data is however limited to daytime retrievals simultaneous to FAAM aircraft flights during the month of June across two years, and so is not sourced from a large enough dataset to be considered a climatology. It is shown here mainly to illustrate the potential of the cloud mask and to suggest certain locations across the Sahara over which clouds most

frequently appear to form. Four main areas of cloud formation are identified using Fig. 8a: A) the area surrounding the Western Saharan town of Smara; B) the El Eglab Massif in western Algeria (~300 m); C) the Adrar plateau in Mauritania (340 m); and D) parts of the remote low-lying desert region in northern Mali.

Marsham et al. (2008), Huang et al. (2010), Cuesta et al. (2009) and Birch et al. (2012) suggest two possible triggers for cloud formation: 1) surface albedo anomalies and 2) topography. Figure 8b shows in each of the areas A to D, there is an

indication that a dark albedo anomaly does indeed exist, attributed in general to local geological features. These will contribute to higher surface temperature, convergence, a deeper SABL and reduced entrainment (Garcia-Carreras et al., 2011), which in turn may be increasing the convective available potential energy, all favouring cloud formation.

The terrain map in Fig. 8c suggests that three of the cloudy areas (A, B and C), are also associated with high ground (relative to the surrounding area). These mountainous regions also have low albedo values, due to their rocky surfaces, making the

distinction between the effects of albedo and those of elevation difficult. However, the area with the highest cloud frequency of the four shown in Fig. 8a is the low lying desert area in Mali (labelled D), which suggests that elevated terrain is not always a critical factor in cloud development.

Overall, these findings broadly show consistency with previous studies (Marsham et al., 2008; Cuesta et al., 2009), although the dataset is not yet comprehensive enough to draw robust conclusions about cloud formation mechanisms. A detailed in-

vestigation into these mechanisms are beyond the scope of this paper, but further study is advocated by the authors into the relationship between albedo, terrain and cloud development in the SABL, by using the SEVIRI cloud mask datasets discussed here.

## 4    Discussion of the possible role of dust in SEVIRI errors

Of the cloud mask threshold tests described in Hocking et al. (2011), we speculate that the *gross* test (usually responsible for

~80 % of detected cloudy pixels) is the test which may be retrieving more cloudy pixels than the aircraft suggests to be present, since if the atmospheric dust loading is high, $T_B$ can be diminished by dust absorption (Haywood et al., 2005). The principle of the gross test is quite simple; cloud tops have a much lower brightness temperature than that of the underlying desert surface (which is simulated by the MetUM and RTTOV). Therefore dust reducing $T_B$ may lead to an instance of incorrectly flagged cloud.

Figure 9 demonstrates this effect using clear-sky pixels as they cross a boundary between adjacent areas of high and low dust loading, and shows a unique dust event which occured during Fennec (Sodemann et al., 2015). In the eastern half of Fig. 9a, the clear-sky $T_B$ value is shown to differ from the MetUM desert skin temperature by ~5 K. At ~9° W, the SEVIRI $T_B$ value sharply decreases, in line with the location of the boundary into the dustier area in Fig. 9b, which is identifiable by the



pink/magenta colors (Brindley et al., 2012) in the RGB imagery. By 13° W, the difference between $T_B$ and the background model skin temperature is in excess of 20 K. This value of 20 K occurs in clear-sky conditions due to dust alone, but this difference will increase further for pixels contaminated by cloud. It is therefore feasible that some of the cloud flagging seen in this study for partially cloud-contaminated (and even cloud-free) pixels may be occurring because of the presence of high dust

loading; in other words, SEVIRI's observed radiance ($R_j^m$) may be modified by dust sufficiently such that the term $R_j^m - R_j^c$ in Eq. 1 (which takes no account of dust in the longwave end of the spectrum) could be large enough that the *gross* test's threshold for identifying a cloud is met.

This effect may explain the lower-than-expected overall value of 68 % ± 5 % obtained by *CloudRetrieval* for aircraft cloud fractions greater than 0.9. If the threshold in the Hocking et al. (2011) *gross* test is met due to a combination of partial cloud

contamination and dust, this would explain why the scheme sometimes flags a cloud when aircraft cloud fractions are low. Since the *gross* test will flag a cloud in *AllCloud* whenever a cloud is flagged in *CloudRetrieval*, this would also explain why the general shape of the distribution in each histogram (Figs. 5 and 6) always appears to be similar for the range 0.1–0.9 . The aircraft cloud fraction range 0.1–0.9 has been included in Table 3 to show each discrete contribution.

In an intercomparison of 12 cloud mask products under the framework of the fourth-annual cloud retrieval evaluation work-

shop (CREW-4), Roebeling et al. (2014) identified the Sahara as one of three areas in the SEVIRI full-disk image which showed the largest discrepancy between the cloud masks (for a specific SEVIRI frame taken in June 2008). The cause of this disagreement was identified as differing thresholds for the detection of thin cirrus. However, in light of the tendency found in this study for SEVIRI to overestimate cloudiness, we propose that perhaps at least some of this disagreement might be related to the effect of dust absorption on whether the different cloud mask algorithms will trigger a cloud flag. Dust was also a factor

in another key region in the CREW-4 study, the Arabian Peninsula, with a dust storm in the area being the suggested cause of the mismatch between the 12 cloud masks. The incorrect flagging of pixels in SEVIRI images by dust has also been suggested by Banks and Brindley (2013).

Dust, combined with low cloud fractions, may also be a key cause for the poorly matched cloud-top height estimates identified in this study. When a cloud is observed by SEVIRI, two of the three Met Office retrieval schemes seem to dominate

over the Sahara; the minimum residual scheme and the profile matching scheme. One hypothesis is that the minimum residual scheme could be incorporating an underestimate of the surface $T_B$ contribution in partially cloudy pixels, due to dust absorption between the surface and the cloud base. Using the dropsonde profile of Fig. 7c, the 15-20 K difference shown in the western half of Fig. 9a would translate to a height error of ~1,500–2,000 m. When cloud contamination is added, this height error would undoubtedly increase.

Hamann et al. (2014) showed that the Met Office CTH retrievals performed comparably to those developed by EUMETSAT, NOAA, Meteo-France, and NASA. Although not the only methods of cloud retrieval, the "radiance fitting" and "radiance ratioing" methods (characterised here by the profile matching and minimum residual schemes respectively) are widely employed by other centres (See Table 4 of Hamann et al. (2014)). Aerosols are not accounted for in the longwave radiative transfer calculations by any of the centres, which implies the possibility that over/underestimates of CTH might be occurring over the Sahara





in many of these products. In summary, with no ground truth to verify CTH in intercomparison studies, it is well possible that at least some of the cirrus observed by satellites over the Sahara is, in actual fact, cumuliform cloud atop the SABL.

### 4.1 Considerations regarding model skin temperatures

Aside from dust, another consideration in locating the source of cloud product errors over the Sahara is the accuracy of the MetUM model skin temperature. Although Fig. 7c shows that large CTH errors can arise despite an NWP profile which is well-matched with observational data, it cannot necessarily be assumed that the model skin temperature is also an accurate reflection of reality. Given the rapidly changing temperatures of the desert surface, especially in the summer, the value of $T_B$ at any given instant can be problematic for the model, leading to another source of potential error. Although Fig. 9 shows that dust can have an effect on the *relative* difference between model skin temperature and the observed SEVIRI $T_B$ at 10.8 μm, errors in the *absolute* value of model skin temperature also have the potential to introduce unknown effects.

### 5 Conclusions

Derived products used to identify and characterise cloud over the Sahara desert based on the SEVIRI instrument aboard the MSG geostationary satellite have been compared to airborne cloud remote sensing products from the FAAM BAe146 aircraft during the Fennec Campaign. This provides the first evaluation of satellite cloud data using detailed observations in the extremely data-sparse region of the Sahara, which has an important impact on the global radiation budget. A novel method for identifying cloud above the aircraft using pyranometer data is described, as well as the methodology for cloud identification below the aircraft using a combination of the lidar and Heimann radiometer.

Two configurations of the Met Office cloud mask product, *AllCloud* and *CloudRetrieval*, are used in the comparison. *AllCloud*, a configuration which is intended to include partially cloud-contaminated pixels, is matched to an aircraft cloud fraction greater than 0.1 in 80 % ± 8 % of the SEVIRI pixels found along the combined BAe146 flight paths. In the case of the *CloudRetrieval* mask, which is not intended to flag partially cloudy SEVIRI pixels, we find that 68 % ± 5 % of the cloudy pixels are matched with aircraft cloud fractions greater than 0.9. The cloud mask products achieve higher accuracy for non-cloudy pixels, with 91 % ± 8 % and 97 % ± 5 % correctly identified pixels in *AllCloud* and *CloudRetrieval* respectively. Some partially-cloudy pixels (aircraft cloud fraction 0.1-0.9) are masked as cloudy in *CloudRetrieval* (19 %), suggesting care must be taken when using *CloudRetrieval* for applications which require completely cloud-free pixels. These results are complemented by a qualitative visualisation of how the satellite and aircraft datasets compare. This is especially important for scenes in which the cloud patterns are broken / cellular in nature, and show that visually speaking, the datasets agree very well.

There is a tendency for SEVIRI to flag cloud more frequently than the aircraft. We speculate that this may be due predominantly to the influence of two key factors: 1) high dust loadings, which reach cloud base (~5–6 km) in the great majority of cases and act to decrease the 'background' brightness temperature and 2) small cloud horizontal extents, as 67 % of clouds were found to be smaller than the SEVIRI resolution (~3 x 3 km$^2$). However, despite these limitations, we believe that both of





the cloud mask configurations discussed here are capable of providing very useful datasets with many potential applications, including NWP model assimilation, operational forecasting, meteorological research, and climate studies.

We present a first estimate of the typical daytime geographical distribution of cloud over the Sahara in June. We identify four areas which seem prone to cloud formation, and show that these areas are consistent with existing theories that surface albedo anomalies and topography may be the key drivers of Saharan cloud formation (Marsham et al., 2008; Cuesta et al., 2009; Huang et al., 2010). The dataset used is too limited to represent a true climatology, but we advocate the use of the cloud masks to create a Saharan cloud climatology in a future publication.

Finally, estimates of cloud-top height from the Met Office's suite of cloud products have been compared with lidar-derived CTH estimates from the BAe146. We find in general that the retrievals differ substantially from the lidar-derived CTH over the Sahara. We have identified a relationship between the effective cloud amount (the product of cloud fraction and cloud emissivity) and errors in the CTH values, and suggest that both cloud horizontal extent and dust loading may again be the key factors which are negatively affecting the results. A SEVIRI-derived CTH, theoretically speaking, might still be useful if the data were to be restricted to using high values of effective cloud amount. However, it is more likely that in order to create a sufficiently reliable dataset, further investigation will be required into the effects of mineral dust, model skin temperatures, and selection criteria for component retrieval schemes in this particular part of the World.

**Acknowledgments**

Airborne data were obtained using the BAe-146-301 Atmospheric Research Aircraft (ARA) flown by Directflight Ltd and managed by the Facility for Airborne Atmospheric Measurements (FAAM), which is a joint entity of the Natural Environment Research Council (NERC) and the Met Office. Fennec was funded by NERC (grant NE/G017166/1), and all those involved in making the campaign a success are warmly thanked. John Kealy's time for this project has been awarded within the framework of the Met Office Chief Scientist internal secondment scheme. Franco Marenco has developed the aircraft cloud mask at the University of Leeds, within the framework of the Met Office Academic Partnership scheme.



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



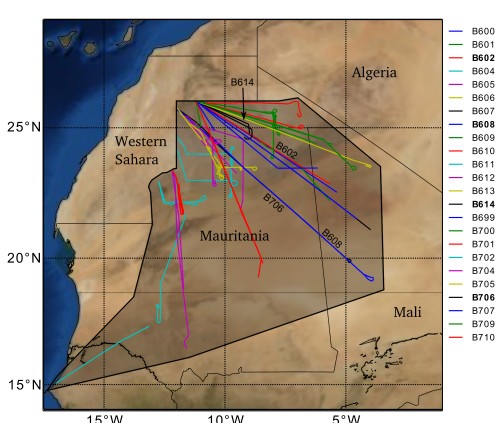

**Figure 1.** Domain of interest in the Sahara desert, with flight tracks of the BAe146 during Fennec. The shaded polygon surrounds the sampling area. Important flights which are analysed in detail throughout the text are highlighted.



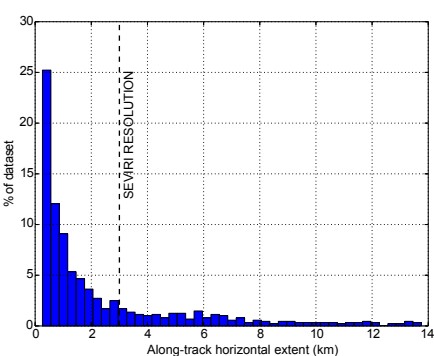

**Figure 2.** Distribution of cloud sizes from the full aircraft dataset. 67 % of the clouds detected by the aircraft exhibit a horizontal extent smaller in size than SEVIRI's resolution.





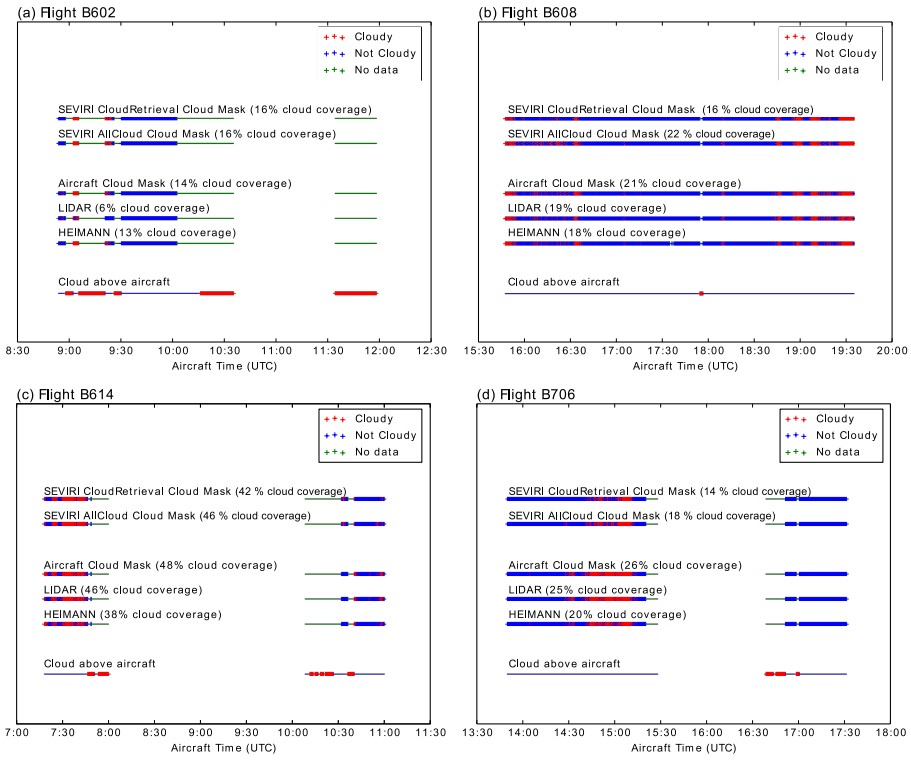

**Figure 3.** Comparison of *CloudRetrieval* (SEVIRI pixels fully cloud contaminated) and *AllCloud* (partially cloud contaminated SEVIRI pixels also included) cloud masks against the aircraft cloud masks for flights **(a)** B602; **(b)** B608; **(c)** B614; and **(d)** B706. Percentages show the amount of cloudy points (red) as a fraction of the total (red+blue) points for which a cloud flag has been assigned. Gaps in the data (green) can be attributed to either the filtering of cloud detected above the aircraft, aircraft altitudes of < 4500 m, or missing points in the raw data. The BBR-derived filter for cloud above the aircraft is also included.



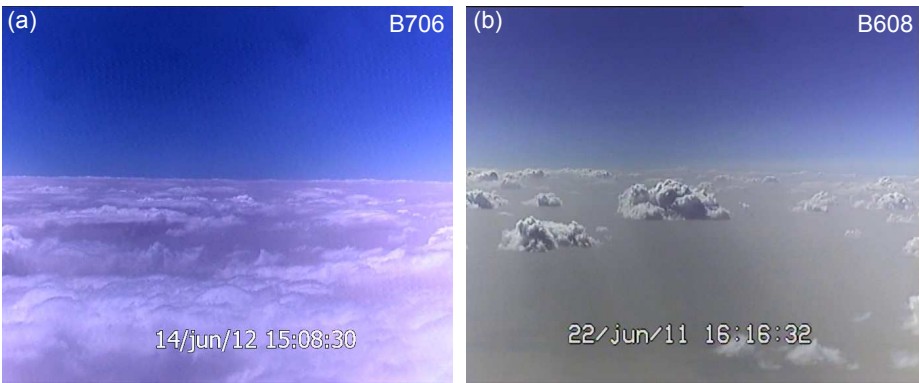

**Figure 4.** Snapshot from the BAe146 rearward-facing digital camera at **(a)** 1508 UTC during B706 and **(b)** 1616 UTC during B608, showing the cloud horizontal extent and structure of the cloud in each scene. Note the strong contrast in (b) between the dusty SABL and the free-atmosphere above, with the clouds lying at this boundary.





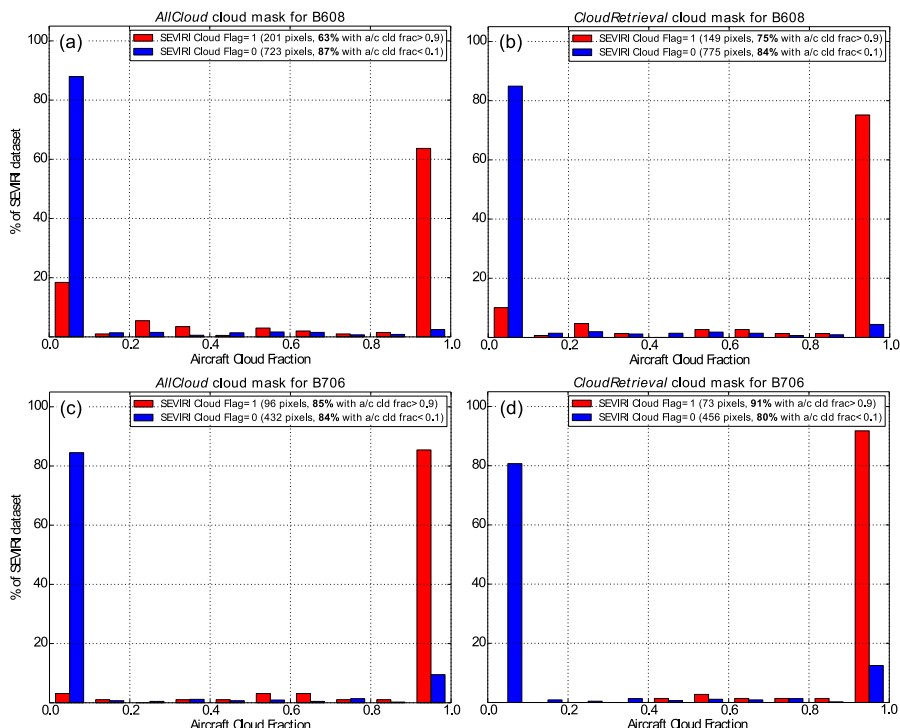

**Figure 5.** Histogram comparison of aircraft cloud fractions as matched to SEVIRI cloud flags showing pixels flagged as cloudy (red) and non-cloudy (blue) in SEVIRI. Percentage values in the legend pertain to aircraft cloud fractions greater than 0.9 for cloudy SEVIRI pixels, and less than 0.1 for non-cloudy SEVIRI pixels, to indicate the proportion of the dataset deemed to match. **Top:** Flight B608 **(a)** *AllCloud* and **(b)** *CloudRetrieval* cloud masks. **Bottom:** Flight B706 **(c)** *AllCloud* and **(d)** *CloudRetrieval* cloud masks.





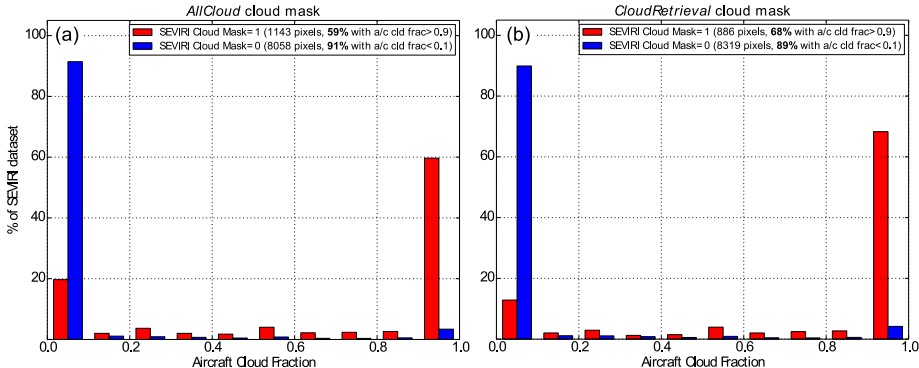

**Figure 6.** Histogram containing the full Fennec dataset to show the statistical agreement between the SEVIRI and aircraft cloud masks, for

**(a)** *AllCloud* and **(b)** *CloudRetrieval*. Histograms are as described in Fig. 5.





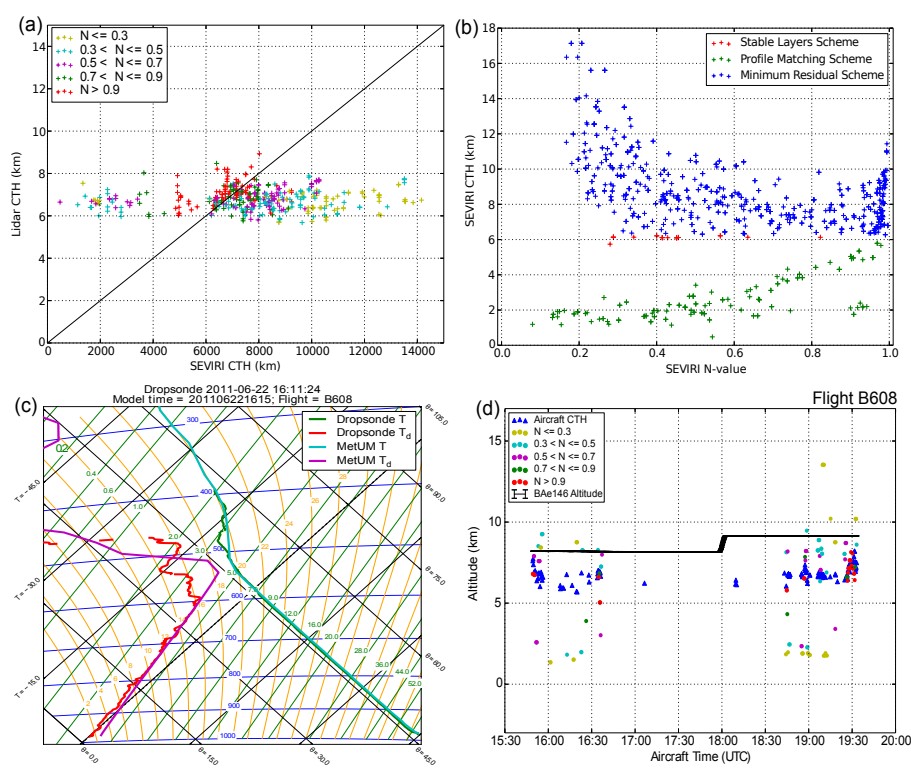

**Figure 7. (a)** Scatterplot of CTH between SEVIRI and the aircraft datasets (all flights, all retrieval schemes) **(b)** SEVIRI CTH retrievals for every pixel associated with a BAe146 data-point, plotted against the SEVIRI $N$ value, and streamed by the type of retrieval scheme used (all flights) **(c)** Thermodynamic diagram of temperature and dewpoint from a dropsonde profile of the SABL from flight B608, overlaid with the nearest MetUM temperature profile **(d)** Time-series of lidar CTH (blue triangles) and SEVIRI CTH retrievals for B608. Pixels where cloud above the aircraft was detected have been omitted in each panel.



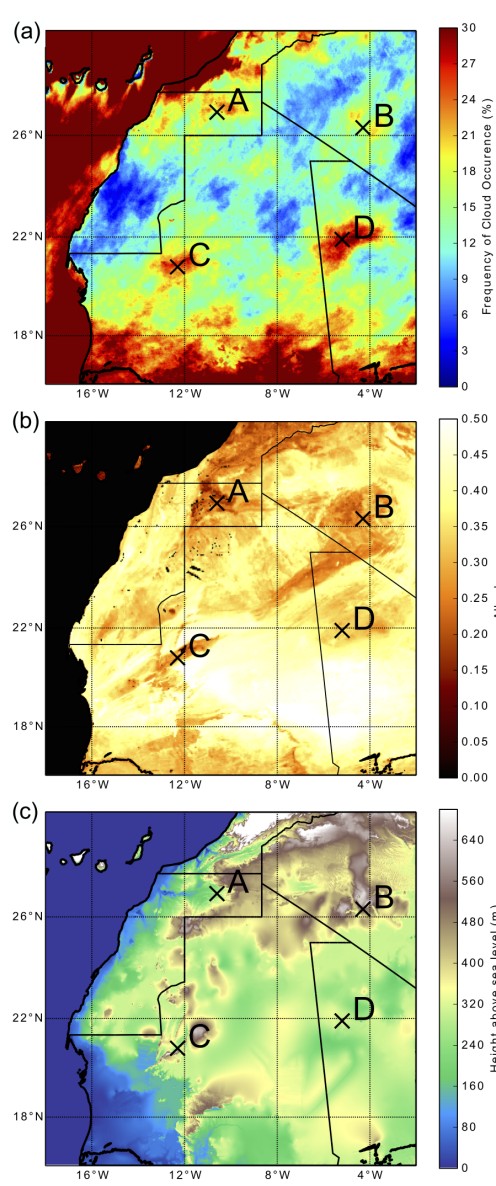

**Figure 8. (a)** Compilation of all cloudy SEVIRI pixels (*CloudRetrieval* mask) across all the Fennec BAe146 flight dates. Lettering (A–D) indicates the cloudiest land-based areas in the region of interest. Image compiled from 670 SEVIRI *CloudRetrieval* cloud mask frames in the daytime for June 2011 and 2012 (simultaneous to research flight times only) **(b)** Surface albedo map of the region. **(c)** Terrain map of the region.





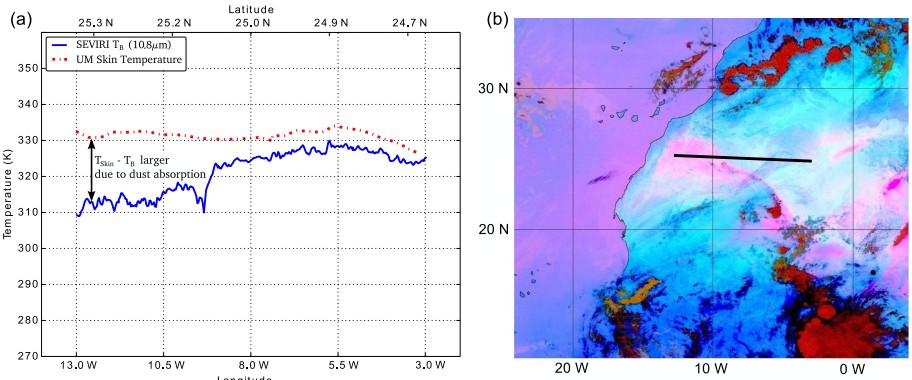

**Figure 9. (a)** Cross section of the Sahara from 20 June 2011 at 1430 UTC, showing how MetUM model skin temperatures (used as an input to the CTH calculations) compare with SEVIRI brightness temperature in the IR. **(b)** Dust RGB showing the location of the cross section in (a) as a black line. Areas of high dust loading show up in bright pink/magenta. The difference between SEVIRI $T_B$ and model $T_{skin}$ is larger in the dusty area to the west, likely due to absorption by the aerosols.



| Date / Time | Flight Number | Percentage of cloud in BAe146 data | Percentage of cloud in SEVIRI data | | Cloud detected above aircraft? (y/n) |
|---|---|---|---|---|---|
| | | | *AllCloud* | *Cloud-Retrieval* | |
| **2011** | | | | | |
| 17 June 07:48-12:41 | B600 | 2 % | 8 % | 5 % | y |
| 17 June 14:43–19:37 | B601 | 32 % | 43 % | 39 % | y |
| 18 June 08:10–12:40 | B602 | 14 % | 16 % | 16 % | y |
| 20 June 12:47–17:51 | B604 | 8 % | 7 % | 3 % | y |
| 21 June 08:10–11:58 | B605 | 7 % | 4 % | 3 % | n |
| 21 June 14:04–19:20 | B606 | 3 % | 3 % | 2 % | n |
| 22 June 08:04–12:37 | B607 | 7 % | 2 % | 2 % | n |
| 22 June 15:10–20:16 | B608 | 21 % | 22 % | 16 % | n |
| 24 June 11:29–16:45 | B609 | 42 % | 47 % | 40 % | y |
| 25 June 07:31–12:17 | B610 | 0 % | 1 % | 0 % | n |
| 25 June 14:14–19:16 | B611 | 3 % | 7 % | 1 % | y |
| 26 June 07:29–12:22 | B612 | 1 % | 0 % | 0 % | n |
| 26 June 13:55–18:59 | B613 | 3 % | 2 % | 1 % | n |
| 27 June 06:34–11:39 | B614 | 48 % | 46 % | 42 % | n |
| **2012** | | | | | |
| 6 June 12:01–16:50 | B699 | 2 % | 5 % | 5 % | y |
| 8 June 07:56–12:57 | B700 | 3 % | 0 % | 0 % | n |
| 9 June 07:55–13:08 | B701 | 0 % | 0 % | 0 % | n |
| 10 June 08:04–12:41 | B702 | 12 % | 3 % | 3 % | y |
| 11 June 12:14–17:19 | B704 | 50 % | 62 % | 60 % | y |
| 12 June 11:27–17:07 | B705 | 1 % | 8 % | 3 % | y |
| 14 June 13:07–18:13 | B706 | 26 % | 18 % | 14 % | n |
| 15 June 09:13–14:33 | B707 | 39 % | 27 % | 14 % | y |
| 17 June 12:14–17:24 | B709 | 6 % | 4 % | 3 % | n |
| 18 June 07:51–13:11 | B710 | 0 % | 0 % | 0 % | n |

**Table 1.** Dates/Times for each BAe146 flight, with the percentage of the dataset which is flagged as cloudy shown for the aircraft, the *AllCloud* SEVIRI mask, and the *CloudRetrieval* SEVIRI mask. Also shown is whether or not cloud above the aircraft was observed by the BBR.



| Flight Number | Cloudy SEVIRI pixels with ACF > 0.1 | Clear-sky SEVIRI pixels with ACF < 0.1 | Ratio of Cloudy SEVIRI pixels to total sample size | Uncertainty |
|---|---|---|---|---|
| | *AllCloud* | *AllCloud* | *AllCloud* | *AllCloud* |
| B600 | 57 % | 100 % | 14 / 183 | ± 14 % |
| B601 | 84 % | 91 % | 106 / 246 | ± 17 % |
| B602 | 79 % | 95 % | 29 / 177 | ± 6 % |
| B604 | 60 % | 89 % | 25 / 366 | ± 8 % |
| B605 | 80 % | 93 % | 15 / 358 | ± 1 % |
| B606 | 63 % | 96 % | 16 / 459 | ± 8 % |
| B607 | 100 % | 91 % | 11 / 689 | ± 1 % |
| B608 | 82 % | 88 % | 201 / 924 | ± 16 % |
| B609 | 84 % | 77 % | 154 / 325 | ± 20 % |
| B610 | 33 % | 99 % | 3 / 370 | ± 1 % |
| B611 | 60 % | 95 % | 15 / 211 | ± 12 % |
| B612 | 0 % | 98 % | 0 / 441 | ± 0 % |
| B613 | 80 % | 96 % | 5 / 296 | ± 7 % |
| B614 | 93 % | 65 % | 95 / 207 | ± 14 % |
| B699 | 50 % | 98 % | 8 / 156 | ± 15 % |
| B700 | 0 % | 96 % | 0 / 252 | ± 1 % |
| B701 | 0 % | 100 % | 0 / 572 | ± 0.2 % |
| B702 | 92 % | 88 % | 12 / 430 | ± 6 % |
| B704 | 81 % | 73 % | 152 / 247 | ± 15 % |
| B705 | 0 % | 98 % | 36 / 451 | ± 12 % |
| B706 | 97 % | 84 % | 96 / 528 | ± 15 % |
| B707 | 80 % | 63 % | 128 / 474 | ± 23 % |
| B709 | 95 % | 93 % | 22 / 570 | ± 4 % |
| B710 | 0 % | 99 % | 0 / 274 | ± 0 % |
| All Flights | 80 % | 91 % | 1143 / 9205 | ±8 % |

**Table 2.** Cloud mask matching statistics for the *AllCloud* SEVIRI cloud mask, sample size for the number of SEVIRI pixels which contain aircraft points, and the temporal measurement uncertainty as described in Sect. 2.5. Aircraft cloud fraction is denoted by ACF.



| Flight Number | Cloudy SEVIRI pixels with ACF > 0.9 | Cloudy SEVIRI pixels with 0.1 < ACF < 0.9 | Clear-sky SEVIRI pixels with ACF < 0.9 | Ratio of Cloudy SEVIRI pixels to total | Uncertainty |
|---|---|---|---|---|---|
| | *CloudRetrieval* | *CloudRetrieval* | *CloudRetrieval* | *CloudRetrieval* | *CloudRetrieval* |
| B600 | 78 % | 11 % | 100 % | 9 / 183 | ± 10 % |
| B601 | 68 % | 24 % | 99 % | 97 / 246 | ± 13 % |
| B602 | 59 % | 21 % | 99 % | 29 / 177 | ± 4 % |
| B604 | 58 % | 17 % | 97 % | 12 / 366 | ± 5 % |
| B605 | 78 % | 22 % | 96 % | 9 / 358 | ± 0.8 % |
| B606 | 20 % | 60 % | 99 % | 10 / 459 | ± 3 % |
| B607 | 100 % | 0 % | 96 % | 10 / 689 | ± 1 % |
| B608 | 75 % | 15 % | 96 % | 149 / 924 | ± 10 % |
| B609 | 69 % | 19 % | 92 % | 131 / 326 | ± 12 % |
| B610 | 0 % | 0 % | 100 % | 0 / 371 | ± 0.5 % |
| B611 | 0 % | 67 % | 100 % | 3 / 212 | ± 3 % |
| B612 | 0 % | 0 % | 99 % | 0 / 441 | ± 0 % |
| B613 | 0 % | 100 % | 99 % | 1 / 296 | ± 2 % |
| B614 | 75 % | 20 % | 98 % | 87 / 207 | ± 10 % |
| B699 | 13 % | 38 % | 99 % | 8 / 156 | ± 4 % |
| B700 | 0 % | 0 % | 96 % | 0 / 252 | ± 0.4 % |
| B701 | 0 % | 0 % | 100 % | 0 / 572 | ± 0 % |
| B702 | 91 % | 0 % | 91 % | 11 / 430 | ± 4 % |
| B704 | 59 % | 21 % | 89 % | 148 / 246 | ± 11 % |
| B705 | 0 % | 0 % | 100 % | 15 / 451 | ± 6 % |
| B706 | 92 % | 8 % | 88 % | 73 / 529 | ± 9 % |
| B707 | 65 % | 21 % | 77 % | 66 / 475 | ± 12 % |
| B709 | 67 % | 33 % | 98 % | 18 / 570 | ± 3 % |
| B710 | 0 % | 0 % | 100 % | 0 / 274 | ± 0 % |
| All Flights | 68 % | 19 % | 97 % | 886 / 9205 | ± 5 % |

**Table 3.** As Table 2 but for the *CloudRetrieval* SEVIRI cloud mask. The ACF range 0.1–0.9 is also shown in this table, as discussed in Sect. 4.