# Peer review of "Clouds over the summertime Sahara: An evaluation of Met Office retrievals from Meteosat Second Generation using airborne remote sensing"

_Atmospheric Chemistry and Physics, 2016_

## Referee Comment (RC1) · Anonymous Referee #2 · 9 Dec 2016

The manuscript "Clouds over the summertime Sahara: an evaluation of the Met Office Meteosat retrievals using airborne remote sensing" of J.C. Kealy et al., submitted to Atmospheric Chemistry and Physics, presents a comparison between the Met Office MSG cloud retrievals and aircraft cloud retrievals (cloud fraction and cloud top height). Comparing satellite observations with aircraft observations is a complicated task since the temporal and spatial scale are very different. But these exercises are very important to highlight the weaknesses of satellite retrieval. Overall, the manuscript is well written and well structured. The authors provide important results and interesting perspectives of this work. Finally, I do recommend this manuscript for publication in ACP but after a minor revision that will answer my comments.

[Figure]

General comments:

1) In the title, it should be mentioned that the Meteosat Second Generation retrieval have been evaluated and not the Meteosat only.

2) It is never mentioned in the manuscript the problem of multi layered clouds for cloud retrieval (especially for cloud top height). I think authors should mention this at least in the minimum residual method weakness.

3) As multi-layered cloud, nothing is provided about surface emissivity. What model did you used in the simulations? what are the error in the CTH retrieval? Surface temperature error is clearly an important source of error but surface emissivity is also important.

4) The Figure 3 is very difficult to read. I suggest authors to find an other way to present the result. 5) The aircraft cloud retrieval is based on different thresholds (i.e., 3K for the radiometer, 7 W.m-2.s-1 for the BBR). What is the sensitivity of the retrievals to those thresholds ? A discussion on that should be added.

Specific comments:

Page 3 Line 30: What is the RTTOV version ?

Page 3 Line 35: the reference to Eyre and Menzel is missing in the references list.

Page 3 Line 41: It should be explain what "error-weighted" means ?

Page 3 Line 59: I am not an English native speaker but it sounds to me that there one "by" in excess.

Page 4 Line 33: what is the spectral resolution and the absolute calibration error of the Heimann radiometer ?

Page 4 Line 55: what is the accuracy of the CTH ?

Page 5 Line 45: It is not clear to me how did you use the MODIS albedo data ? The

part need more description.

Page 5 Line 50: It has been shown by EUMETSAT that MSG Level 1.5 data has a constant geo-referencing offset towards the North and the West direction of 1.5 km. How did you take into account this ? Some words about that should be included.
* * *

---

## Referee Comment (RC2) · Anonymous Referee #1 · 19 Jan 2017

This paper attempts to evaluate the performance of the UK Met Office cloud mask and cloud top height (CTH) retrievals applied to the Meteosat SEVIRI geostationary imager over the Sahara. The evaluation relies on lidar and radiometer observations obtained from the BAe-146 aircraft flown from Fuerteventura as part of the Fennec program in June 2011 and 2012. The authors show good agreement (or at least explainable comparisons) between the SEVIRI derived cloud detection and CTH and that retrieved by the aircraft instruments, and include a first-look Saharan cloud fraction "climatology" as well as a brief foray into the mechanisms that drive cloud formation in the region.

The text is exceptionally well written, the results are clearly presented, and I find no significant deficiencies in the analysis. I therefore recommend the paper for publication

after only minor revisions.

General Comments

The authors enter somewhat treacherous territory when attempting to evaluate a satellite derived cloud mask, for which the answer to "What is a cloud?" is often "I know a cloud when I see it." As they rightly point out, the design of a cloud mask, and the "accuracy" of cloud detection and derived cloud fractions, are determined in part by the science questions asked, e.g., detecting/removing clouds for a clear sky retrieval product vs detecting clouds for a cloud retrieval product. This in addition to the spectral channel information, sensor spatial resolution, etc. Many investigations lack an appropriate level of consideration for these distinctions, but the authors do a nice job here. My only quibble with the cloud mask analysis (and it is indeed only nit-picking) is the use at times of the term "validation," which implies a comparison of a given retrieved parameter with the direct-measured truth. I would suggest using the term "evaluation" as is done in the title and abstract, in particular because a satellite derived cloud mask is an ill-defined parameter and the fact that the "truth" used here, from the lidar and radiometer, are in fact retrievals themselves.

The authors are on more solid ground with the CTH evaluation, though I have a concern with the analysis as presented. The authors acknowledge that partly cloudy pixels that are treated as overcast will often yield biased CTH retrievals, and they include a nice discussion of the mechanisms for these biases. However, in Fig. 7 they use the SEVIRI derived effective cloud amount N to show the relationship between sub-pixel cloudiness and cloud top biases instead of the aircraft derived cloud fractions. The authors themselves acknowledge that N is not explicitly calculated and should be used with caution. I suggest they either re-create this figure using the SEVIRI pixel-level aircraft cloud fractions instead of N, or add a figure/panel showing CTH biases as a function of the aircraft cloud fractions like what was done for the cloud mask analysis in Figs. 5 and 6. I believe this would be a much more defensible approach.

Specific Comments

p. 3, line 14: I assume the Hocking (2011) cloud mask is a widely-used product at the Met Office?

p. 4, Eqns. 1, 2: Should N be a function of cloud top pressure p?

p. 4, line 17: How is the channel variance defined?

p. 5, line 32: Can the authors comment on the size of the across-track field of view?

p. 6, lines 9-12: How frequent are these missed detections?

Section 2.2: I assume a down-viewing imager that could be co-located with the lidar and radiometer was not flown? This would have been useful for evaluating the lidar and radiometer cloud masks.

p. 7, line 10: Why are above-aircraft cloud detections not useful for the cloud mask comparison?

p. 7, lines 21-25: The across-track FOV of the aircraft is obviously not as wide as a SEVIRI pixel, so the aircraft derived cloud fractions do not sample the entire SEVIRI pixel. Can the authors comment on the impacts of this?

p. 7, lines 29-30: Can cloud movement cause an overestimation of the SEVIRI cloud mask uncertainty? As defined the uncertainty implies the assumption that changes in pixel-level cloud mask are due to cloud formation/dissipation.

p. 9, lines 19-25: Can the authors comment on the role of SEVIRI cloud mask "false positives" in regards to the positive cloud mask results having aircraft cloud fractions below 0.1?

Fig. 7a: The x-axis label states the units as (km), however the tick labels appear to have units (m).

Fig. 9b: What spectral channels are used to create this RGB?

Section 4.1: Could cloud mask false positives also be due to solar reflectance tests?

---

## Author Comment (AC1) · 24 Feb 2017

**Clouds over the summertime Sahara: An evaluation of Met Office retrievals from Meteosat Second Generation using airborne remote sensing**

**Author response to reviewer comments**

February 24, 2017

We'd like to thank both reviewers for their words of appreciation for our work, and for their detailed review. In what follows, we address the reviewer #2 comments and provide a detailed response. We believe that the reviewer comments help us to improve the manuscript, and bring it to publication standard.

Please note that in our responses, page and line numbers now refer to the revised manuscript, which we will submit as soon as we are requested to do so by the journal.

**2. Comments from Reviewer #2**

**2.1 General comments**

**1) Referee comment:**

In the title, it should be mentioned that the Meteosat Second Generation retrievals have been evaluated and not the Meteosat only.

**Author's response:**

We have opted to update the title to help incorporate this.

**Author's changes in manuscript:**

We have retitled the manuscript to "Clouds over the summertime Sahara: An evaluation of Met Office retrievals from Meteosat Second Generation using airborne remote sensing"

The abstract (Page 1 Line 4) now refers to Meteosat Second Generation, rather than Meteosat. We have also changed "Meteosat" to "MSG" at Page 3 Line 6 and Page 3 Line 8 in the revised manuscript.

**2) Referee comment:**

It is never mentioned in the manuscript the problem of multi layered clouds for cloud retrieval (especially for cloud top height). I think authors should mention this at least in the minimum residual method weakness.

**Author's response:**

This is a very valid point, and does deserve mention. The minimum residual method assumes a single layer of cloud, however, some 20 – 40 % of cloud scenes globally have multiple layers of cloud, and this can result in retrieved cloud top heights in multi-layer scenes being intermediate between the two layers if the higher cloud is optically thin. On the other hand, we believe that for the present research, in most cases only one cloud layer is present for each of the scenes. We believe this to be reasonable because of the mechanisms for cloud formation discussed in Cuesta

et al. (2009), i.e. humidities being too low in the lower boundary layer for a low cloud layer, so that boundary layer clouds can only form near the capping inversion at about 6,000 m. The lidar curtains obtained on the aircraft also do not show evidence of distinct separate cloud layers at more than one level. Finally, cases with clouds above the aircraft (flying at ~ 8,000 m) have been screened out as described in section 2.3.

**Author's changes in manuscript:**

We add the following in Section 2.1 (Page 5 Line 26-32 in the revised manuscript): "Finally, we note that in general the cloud-top height product will sample multi-layered cloud for a significant portion (~20-40 %) of the SEVIRI full-disk. This can sometimes have the effect of cloud-top heights in multi-layer scenes appearing at an intermediate level for pixels retrieved with the minimum residual scheme, if the higher layer is optically thin. However, for the present dataset we are confident that the assumption of a single cloud layer is reasonable because: (1) the boundary layer is very dry as discussed in section 3.3 so that the formation of a lower cloud is very unlikely (Cuesta et al., 2009); (2) the inspection of the lidar curtains (not shown here) does not show any evidence of lower cloud layers, and (3) cases with clouds higher than the aircraft are filtered out, as discussed in section 2.3 below."

**3) Referee comment:**

As multi-layered cloud, nothing is provided about surface emissivity. What model did you used in the simulations? what are the error in the CTH retrieval? Surface temperature error is clearly an important source of error but surface emissivity is also important.

**Author's response:**

Regarding surface emissivity in RTTOV: For land surfaces the University of Wisconsin "UWiremis" IR emissivity atlas is used (Borbas and Ruston, 2010). This provides monthly climatological land surface emissivity values at wavelengths in the range 3.7 m – 14.3 m at a resolution of 0:1  0:1 degrees Lat/Lon, derived from MODIS observations. In the Met Office cloud products suite, the atlas emissivities are averaged over each model grid box for input to the RTTOV simulations. Errors in assumed emissivity will naturally propagate through to the CTH retrievals, and we acknowledge that this may contribute to the errors in an unknown way. We add an acknowledgment of this into Sect. 4.1.

Ref: Borbas, E. E. and B. C. Ruston, 2010. The RTTOV UWiremis IR land surface emissivity module. NWP SAF report. http://nwpsaf.eu/vs_reports/nwpsaf-mo-vs-042.pdf

**Author's changes in manuscript:**

We add the following at Page 4 Line 15-17 of the revised manuscript:  "Land-based surface emissivity values are incorporated into this model using the 'UWiremis' climatological IR emissivity atlas (Borbas and Ruston, 2010), averaged over each model grid box."

We add the the following at Page 15 Line 15-16 of the revised manuscript: "Any error in the assumed surface emissivity in RTTOV would also have a similar repercussion."

We have also added the above reference to the revised manuscript at Page 18 Line 16-17.

**4) Referee comment:**

The Figure 3 is very difficult to read. I suggest authors to find an other way to present the result.

**Author's response:**

We agree with this comment and we thank the reviewer for pointing this out. We attach to this response a revised version of Fig. 3.

**Author's changes in manuscript:**

The revised Fig. 3 is included as Fig. 1 of the attachment.

**5) Referee comment:**

The aircraft cloud retrieval is based on different thresholds (i.e., 3K for the radiometer, 7 W.m-2.s-1 for the BBR). What is the sensitivity of the retrievals to those thresholds? A discussion on that should be added.

**Author's response:**

We have evaluated this sensitivity by changing the 3 K threshold to either 2 K or 4 K for the Heimann test, and this affects the Heimann cloud flag for < 5% of the pixels in the Fennec dataset. A similar test was run for the BBR threshold, and we find that changing the threshold by 1 $Wm^{-2}s^{-1}$ results in a change of ~3% of points flagged as cloudy in the filter.

**Author's changes in manuscript:**

We add the following at Page 7 Line 14-15 of the revised manuscript: "Sensitivity testing on this threshold shows that a change of ±1 K affects the cloud flag for < 5 % of the pixels in the Fennec dataset."

We add the following at Page 8 Line 4-5 of the revised manuscript: "A sensitivity test reveals that a change of this threshold by ±1 $Wm^{-2}s^{-1}$ alters the assignment of cloudy datapoints by ~3 %."

**2.2 Specific comments**

**Referee comment:**

Page 3 Line 30: What is the RTTOV version?

**Author's response:**

The RTTOV version is v11.

**Author's changes in manuscript:**

We add the RTTOV version number to the revised manuscript at Page 4 Line 15.

**Referee comment:**

Page 3 Line 35: the reference to Eyre and Menzel is missing in the references list.

**Author's response:**

This missing reference has been added.

**Author's changes in manuscript:**

Reference added to Page 18 Line 35 in the revised manuscript

**Referee comment:**

Page 3 Line 41: It should be explain what "error-weighted" means?

**Author's response:**

"Error-weighted" is explained by the use of the variance, σ, in Eq. 1. Error-weighting means that each SEVIRI channel is given a different weight depending on the assumed accuracy of the observation. This is dealt with in the equation by dividing by the variance for each channel.

**Author's changes in manuscript:**

We add the following at Page 4 Line 21-23 in the revised manuscript: "The meaning of 'error-weighting' in this context is that each SEVIRI channel is given a different weight depending on the assumed accuracy of the observation, represented by dividing by $\sigma^2_j$ for each channel in Eq. 1."

**Referee comment:**

Page 3 Line 59: I am not an English native speaker but it sounds to me that there one "by" in excess.

**Author's response:**

The extra "by" has been removed.

**Author's changes in manuscript:**

Revised manuscript: Page 5 Line 7

**Referee comment:**

Page 4 Line 33: what is the spectral resolution and the absolute calibration error of the Heimann radiometer?

**Author's response:**

The Heimann is actually broadband (8-14 m), and does not have a "spectral resolution", but we can see why this was misunderstood. When we say that the dataset is "high-resolution" on Page 5 Line 29 of the original manuscript, we actually mean "high temporal resolution", not "high spectral resolution". We have attempted to clarify this in the edit shown below. As for the calibration, the Heimann is calibrated in the laboratory using known targets. The calibration is considered stable.

**Author's changes in manuscript:**

Page 6 Line 19-20 of the revised manuscript now reads: "...the Heimann radiometer measures **broadband** upwelling radiation at a temporal frequency of 1 Hz, providing a high **temporal** resolution dataset..."

We add the following at Page 6 Line 21-22 in the revised manuscript: "This instrument is calibrated in the laboratory using known targets."

**Referee comment:**

Page 4 Line 55: what is the accuracy of the CTH?

**Author's response:**

By inspection of the lidar curtain plots, we estimate the uncertainty of the lidar CTH to be approximately ±150 m, although this has not been independently validated using an alternative measurement; on such a short vertical scale the exact definition of CTH is in any case ambiguous and we refer here to the region with a large scattering coefficient near the cloud top.

**Author's changes in manuscript:**

Page 6 Line 30-31 in the revised manuscript now reads: "The cloud-top range $R_c$ is then converted to CTH, which we estimate to be accurate to within ±150 m."

**Referee comment:**

Page 5 Line 45: It is not clear to me how did you use the MODIS albedo data? The part need more description.

**Author's response:**

MODIS wasn't directly used in this study; It is mentioned here as the source data from which the surface albedo map was created. This albedo map is compared to terrain and cloud distribution (using the SEVIRI cloud mask) in Fig. 8 (Fig. 9 in the revised manuscript). The albedo map is Fig. 8b (Fig. 9b of the revised manuscript), and is described in Sect. 3.4. We have re-worded the end of Sect. 2.4 in an attempt to be more clear.

**Author's changes in manuscript:**

Page 8 Line 13-15 in the revised manuscript now reads: "Topography was provided using data from the GLOBE digital elevation model (Payne et al., 1999). A map of surface albedo, created by a compilation of MODIS satellite data (Gao et al., 2005), was also employed."

**Referee comment:**

Page 5 Line 50: It has been shown by EUMETSAT that MSG Level 1.5 data has a constant geo-referencing offset towards the North and the West direction of 1.5 km. How did you take into account this? Some words about that should be included.

**Author's response:**

We are aware of a constant geo-referencing offset of 1.5 km to the North and West in the SEVIRI level 1.5 data which we have taken into account when evaluating the latitudes and longitudes of each SEVIRI pixel.

**Author's changes in manuscript:**

We add the following at Page 3 Line 12-14 in the revised manuscript: "A constant geo-referencing offset of 1.5 km to the North and West in SEVIRI level 1.5 data has been identified by EUMETSAT, and this has been corrected for in Lat/Lon grids used in this study."

**Additional Author's changes to the manuscript**

Further discussions between the co-authors initated by the helpful reviewer comments have also lead to the following minor changes to the manuscript.

We have added the following to the abstract at Page 1 Line 2 of the revised manuscript: "Novel methods of cloud detection are applied to **airborne remote sensing observations from** the unique Fennec aircraft dataset..."

We have added the following to the abstract at Page 1 Lines 9-12 of the revised manuscript: "The mean cloud field, derived from the satellite cloud mask acquired during the Fennec flights, shows that areas of high surface albedo and orography are preferred sites for Saharan cloud cover, consistent with published theories."

We have added to the following to the abstract at Page 1 Lines 13-14 of the revised manuscript: "The results of the CTH analysis presented here may also have wider implications for the techniques employed by other satellite applications facilities across the World."

We add the following reference at Page 3 Line 3 of the revised manuscript:

Ref: Trzeciak, T., Garcia Carreras, L., and Marsham, J. H.: Cross Saharan transport of water vapour via recycled cold-pool outflows from moist convection, Geophysical Research Letters, 2016.

We add the following to the acknowledgements, at Page 17 Line 2-3 of the revised manuscript: "John Marsham was funded by the NERC project SWAMMA (NE/L005352/1)."

We add the following at Page 16 Line 27-30 of the revised manuscript: "Since the the methods used by the Met Office in the determination of CTH are also applied in other forms by other satellite applications facilities globally, the relevance of the results presented here are not limited to Met Office products, but may also have implications for other cloud-retrieval algorithms which employ similar techniques."

---

## Author Comment (AC2) · 24 Feb 2017

**Clouds over the summertime Sahara: An evaluation of Met Office retrievals from Meteosat Second Generation using airborne remote sensing**

**Author response to reviewer comments**

February 24, 2017

We'd like to thank both reviewers for their words of appreciation for our work, and for their detailed review. In what follows, we address the reviewer #1 comments and provide a detailed response. We believe that the reviewer comments help us to improve the manuscript, and bring it to publication standard.

Please note that in our responses, page and line numbers now refer to the revised manuscript, which we will submit as soon as we are requested to do so by the journal.

**1. Comments from Reviewer #1**

**1.1 General Comments**

**Referee comment:**

1) The authors enter somewhat treacherous territory when attempting to evaluate a satellite derived cloud mask, for which the answer to "What is a cloud?" is often "I know a cloud when I see it." As they rightly point out, the design of a cloud mask, and the "accuracy" of cloud detection and derived cloud fractions, are determined in part by the science questions asked, e.g., detecting/removing clouds for a clear sky retrieval product vs detecting clouds for a cloud retrieval product. This in addition to the spectral channel information, sensor spatial resolution, etc. Many investigations lack an appropriate level of consideration for these distinctions, but the authors do a nice job here. My only quibble with the cloud mask analysis (and it is indeed only nit-picking) is the use at times of the term "validation," which implies a comparison of a given retrieved parameter with the direct-measured truth. I would suggest using the term "evaluation" as is done in the title and abstract, in particular because a satellite derived cloud mask is an ill-defined parameter and the fact that the "truth" used here, from the lidar and radiometer, are in fact retrievals themselves.

**Author's response:**

This is a nice point, and we absolutely agree. We have no problem with using the term "evaluation" in place of "validation".

**Author's changes in manuscript:**

We have replaced "validate" with "evaluate" at Page 2 Line 28 and Page 3 Line 6 of the revised manuscript.

**Referee comment:**

2) The authors are on more solid ground with the CTH evaluation, though I have a concern with the analysis as presented. The authors acknowledge that partly cloudy pixels that are treated as overcast will often yield biased CTH retrievals, and they include a nice discussion of the mechanisms for these biases. However, in Fig. 7 they use the SEVIRI derived effective cloud amount N to show the relationship between sub-pixel cloudiness and cloud top biases instead of

the aircraft derived cloud fractions. The authors themselves acknowledge that N is not explicitly calculated and should be used with caution. I suggest they either re-create this figure using the SEVIRI pixel-level aircraft cloud fractions instead of N, or add a figure/panel showing CTH biases as a function of the aircraft cloud fractions like what was done for the cloud mask analysis in Figs. 5 and 6. I believe this would be a much more defensible approach.

**Author's response:**

This is absolutely true. After some discussion, we have elected to follow the reviewer's suggestion and include this figure for completeness. We attach to this response a copy of this new figure (Fig. 1 in the attachment). However, we do wish to discuss the reason why we did not include this in the original manuscript.

Firstly, as you can see, the plot is very messy, and it is difficult to discern any quantifiable information from it. There do not appear to be any particular trends or biases that relate the absolute cloud-top height to the aircraft cloud fraction. This result simply tells us what we already know; the SEVIRI CTH does not match well with aircraft observations. This is already clear from Fig. 7a (Now Fig. 7 in the revised manuscript), and is also the reason why we chose an individual case study (Flight B608) to try to communicate the nature of the CTH error.

In our opinion, the plot of SEVIRI CTH vs N presents more useful information than the CTH vs aircraft cloud fraction plot. By using the SEVIRI N-value instead of the aircraft cloud fraction, some structure becomes apparent in the SEVIRI data, giving more insight as to the source of the error. This is certainly not ideal however, and we acknowledge this.

Finally, with respect to N not being explicitly calculated, do bear in mind that this is true only for the profile matching scheme; the N-value is explicitly calculated in the minimum residual and stable layer schemes. (See Page 12 Line 24-27 in the revised manuscript.)

**Author's changes in manuscript:**

We have now added a new figure of SEVIRI CTH vs aircraft cloud fraction, which is Fig. 8a of the revised manuscript. Fig. 7a in the original manuscript is now Fig. 7 in the revised manuscript. All subsequent figures are now incremented upwards, giving the manuscript a new total of 10 figures.

We add the following to Page 11 Line 19-29 in the revised manuscript (Fig. 8 now refers to the Fig. 7 of the original manuscript.) "To understand the CTH errors, we split the MSG retrievals into the component schemes from which they are calculated, plotted as a function of the aircraft cloud fraction (Fig. 8a) and effective cloud amount, N (Fig. 8b). Fig. 8a shows us that there is no dependence of retrieved CTH on actual cloud fraction, whereas Fig. 8b shows that there is a strong dependence of both the retrieval method and the retrieved effective cloud amount, and this highlights a retrieval problem under certain circumstances. By presenting the data as in Fig. 8b, it becomes apparent that a relationship exists between the highest/lowest retrieved SEVIRI CTH's and low values of N . This implies that the errors in CTH are intimately connected to the calculation of N . The scheme that shows the best agreement is the "stable layers", in which the CTH is retrieved independently from N and the latter is only computed successively. The "minimum residual" scheme, instead, derives N and the CTH simultaneously, with the results shown in Fig. 8b. Finally, in the profile matching scheme N is not used, and the value reported is the one derived from the "minimum residual"; the fact that there still remains a dependence tells us therefore that there are issues common to both schemes taking place at the same time."

We add the following to Page 16 Line 22-27 of the revised manuscript: "The results of the present research will be used to develop a newer version of the CTH product. The "stable layers" scheme, which mainly makes use of the model thermodynamic profile to identify CTH, is the one giving better results. This seems to be an indication that, for the conditions encountered in this research (i.e. deep and well-defined boundary layer in the daytime summertime Sahara) the model is actually a more reliable source of information for determining CTH than satellite radiances. On the other hand, only a few retrievals make use of this scheme; this seems to indicate that in a future revised version of the product this balance will probably have to be revisited."

**1.2 Specific Comments**

**Referee comment:**

p. 3, line 14: I assume the Hocking (2011) cloud mask is a widely-used product at the Met Office?

**Author's response:**

The cloud mask is used to initiate the cloud retrievals (CTH and effective cloud amount, cloud optical thickness, effective radius and aircraft icing potential). The cloud mask is also used as part of the cloud assimilation into the UKV configuration of the Unified Model. All of these products are available to operational meteorologists as well. So yes, it would be fair to say the mask is widely used at the Met Office, and we are happy to mention this.

**Author's changes in manuscript:**

We add the following at Page 3 Line 22-23 of the revised manuscript: "This mask is widely used at the Met Office for the derivation and assimilation of a series of products, and is derived by applying a variety of..."

**Referee comment:**

p. 4, Eqns. 1, 2: Should N be a function of cloud top pressure p?

**Author's response:**

Yes, N is fundamentally formulated as a function of cloud-top pressure p and we thank the reviewer for pointing out that this may have been unclear in the previous version. We have updated the manuscript to show explicit dependence on p in the equations.

**Author's changes in manuscript:**

Equations 1 and 2: explicit dependence on p now shown.

**Referee comment:**

p. 4, line 17: How is the channel variance defined?

**Author's response:**

We have added details about the channel variance into the manuscript as specified below.

**Author's changes in manuscript:**

We add the following at Page 4 Line 23-27 in the revised manuscript: "The variances are fixed, and equal to 1.23 K, 1.25 K, and 0.57 K for each of the channels used, which are centred at the 10.8 µm, 12.0 µm, and 13.4 µm channels respectively. These R-matrix (observation error) values are a combination of two sources: (a) the measurement "noise" (i.e. instrumental error, from EUMETSAT-published radiometric error data), and (b) the error in the simulated brightness temperatures, derived from off-line O-B monitoring statistics, which represent any errors in the background NWP profiles and the radiative transfer."

**Referee comment:**

p. 5, line 32: Can the authors comment on the size of the across-track field of view?

**Author's response:**

The lidar field of view is 4 milli-radians, meaning that the footprint at sea surface from 8,000 m altitude is ~32 m and at cloud-level (assuming a cloud ~2,000 m below aircraft) is ~8 m. The field of view for the Heimann is much larger, 22°, as specified in the revised manuscript at Page 6 Line 14. This implies a footprint at sea level of ~3 km when flying at ~8,000 m altitude. At cloud level (~2,000 m below aircraft) this is ~800 m.

**Author's changes in manuscript:**

We add the following at Page 6 Line 14-15 in the revised manuscript: "The lidar field of view is 4 mrad, meaning that the footprint at sea surface from 8,000 m altitude is ~32 m and at cloud-level (assuming a cloud ~2,000 m below aircraft) is ~8 m."

We add the following at Page 6 Line 20-21 in the revised manuscript: "This implies a footprint at sea level of ~3 km when flying at 8,000 m altitude, with a footprint of of ~800 m at cloud level (assuming a cloud level ~2,000 m below the aircraft)."

**Referee comment:**

p. 6, lines 9-12: How frequent are these missed detections?

**Author's response:**

The average number of missed detections by lidar across all Fennec flights, expressed a percentage of all cloudy detections, is 15.4%. This pertains to flights for which the overall ratio of cloudy to non-cloudy points in the final cloud mask is greater than 10 % (as was done in the calculation of measurement uncertainty in Sect. 2.5). For reference, when we include flights with cloud amounts less than 10 % as well, this figure rises to 18.3 %. This shows that the Heimann radiometer data is a valuable tool in creating a robust cloud mask product.

**Author's changes in manuscript:**

We add the following at Page 6 Line 33-34 in the revised manuscript:  "We estimate that these missed detections represent ~15 % of all cloudy pixels for days exceeding 10 % cloud coverage."

**Referee comment:**

Section 2.2: I assume a down-viewing imager that could be co-located with the lidar and radiometer was not flown? This would have been useful for evaluating the lidar and radiometer cloud masks.

**Author's response:**

The BAe-146 is fitted with a downward-facing camera, but the images were unfortunately unusable because the high dust loading in the air caused a build up of dust on the lens. This was also the case for the forward and upward facing cameras. This is the reason that only the rearward facing camera was able to produce usable images (Fig. 4). We did spend a significant amount of time viewing this data, but even if the image quality was better, it would have been far too time consuming to give anything more than a qualitative picture of specific situations, as we have done in Fig. 4.

**Author's changes in manuscript:**

None

**Referee comment:**

p. 7, line 10: Why are above-aircraft cloud detections not useful for the cloud mask comparison?

**Author's response:**

Please bear in mind that both the lidar and the Heimann radiometer are both nadir (downward) facing. It is these instruments that are the source of the aircraft cloud mask, which is a well-tested and robust product. Therefore it is very possible for SEVIRI to observe a cloud which lies above the aircraft that these nadir facing instruments should not be able to see. The pyranometer, by contrast, is upward facing but the methods employed to create the pyranometer cloud filter are not the same as those used to create the aircraft cloud mask. The pyranometer cloud filter was designed to remove complete sections of the dataset, as opposed to identifying whether each individual datapoint is cloudy or not cloudy. This is visualised in Figure S2 of the supplement.

**Author's changes in manuscript:**

None

**Referee comment:**

p. 7, lines 21-25: The across-track FOV of the aircraft is obviously not as wide as a SEVIRI pixel, so the aircraft derived cloud fractions do not sample the entire SEVIRI pixel. Can the authors comment on the impacts of this?

**Author's response:**

This is an unfortunate limitation of the technique, and there is little that can be done to gain information about cloud which lies parallel to the aircraft flight track. On Page 9 Line 22-24 of the original manuscript we already commented on this. However, as this effect is random in its nature, it is assumed that overall, on a wide dataset such as the one discussed here, errors can be reasonably assumed to cancel out.

**Author's changes in manuscript:**

We add the following at Page 10 Line 18-19 in the revised manuscript: "As this effect is random in its nature, it is assumed that overall, on a wide dataset such as the one discussed here, errors can be reasonably assumed to cancel out."

**Referee comment:**

p. 7, lines 29-30: Can cloud movement cause an overestimation of the SEVIRI cloud mask uncertainty? As defined the uncertainty implies the assumption that changes in pixel-level cloud mask are due to cloud formation/dissipation.

**Author's response:**

What we have done here is to literally quantify the changes in the cloud mask field; i.e. which pixels change from cloudy to not-cloudy and vice versa between successive frames. This should take into account clouds forming, dissipating, or moving. Indeed, in some cases (see Fig. S4 and Page 8 Line 24 of the original manuscript) we find that the cloud boundaries tend to show the most change, and this may suggest that a significant part of the change will in fact be due to cloud movement.

**Author's changes in manuscript:**

For clarification, we add the following at Page 8 Line 22-23 in the revised manuscript: "Such changes may be the result of cloud formation, dissipation, or movement."

**Referee comment:**

p. 9, lines 19-25: Can the authors comment on the role of SEVIRI cloud mask "false positives" in regards to the positive cloud mask results having aircraft cloud fractions below 0.1?

**Author's response:**

We are not sure what the reviewer is referring to here. Our manuscript already comments on these "false positives". See e.g. on Page 9 Lines 19-25 of the original manuscript. In this paragraph we offer some explanations , including the measurement uncertainty induced by a changing cloud field, the limited cloud horizontal extent, and the possible mis-detection of cloud by the aircraft due to its smaller pixel size.

**Author's changes in manuscript:**

None

**Referee comment:**

Fig. 7a: The x-axis label states the units as (km), however the tick labels appear to have units (m).

**Author's response:**

This has been corrected.

**Author's changes in manuscript:**

Units in this figure are now all set to km. This is now Fig. 7 in the revised manuscript.

**Referee comment:**

Fig. 9b: What spectral channels are used to create this RGB?

**Author's response:**

The RGB is based on the following:

Red: BT12µm -BT10.8µm, Green: BT10.8µm –BT8.7µm, Blue: BT10.8µm, where BT denotes brightness temperature at the given wavelength. The thresholds are then applied to a colour scale.

Further details of this RGB can be found in Brindley et al. (2012), as cited at Page 13 Line 1 in the original manuscript. A catalogue of this product is also available on the EUMETSAT website at

http://oiswww.eumetsat.org/IPPS/html/MSG/RGB/DUST/WESTERNAFRICA/

**Author's changes in manuscript:**

None

**Referee comment:**

Section 4.1: Could cloud mask false positives also be due to solar reflectance tests?

**Author's response:**

Solar reflectance is one of the tests that goes in the cloud mask, therefore the answer to the question is "yes, this is a possibility". However, we have already thoroughly commented on the cloud mask in Section 2.1. Section 4.1 pertains to the uncertainty on the cloud top height product, which is a different thing. We are therefore unsure why the reviewer inserted this remark at this point in the manuscript.

**Author's changes in manuscript:**

None

**Additional Author's changes to the manuscript**

Further discussions between the co-authors initated by the helpful reviewer comments have also lead to the following minor changes to the manuscript.

We have added the following to the abstract at Page 1 Line 2 of the revised manuscript: "Novel methods of cloud detection are applied to **airborne remote sensing observations from** the unique Fennec aircraft dataset..."

We have added the following to the abstract at Page 1 Lines 9-12 of the revised manuscript: "The mean cloud field, derived from the satellite cloud mask acquired during the Fennec flights, shows that areas of high surface albedo and orography are preferred sites for Saharan cloud cover, consistent with published theories."

We have added to the following to the abstract at Page 1 Lines 13-14 of the revised manuscript: "The results of the CTH analysis presented here may also have wider implications for the techniques employed by other satellite applications facilities across the World."

We add the following reference at Page 3 Line 3 of the revised manuscript:

Ref: Trzeciak, T., Garcia Carreras, L., and Marsham, J. H.: Cross Saharan transport of water vapour via recycled cold-pool outflows from moist convection, Geophysical Research Letters, 2016.

We add the following to the acknowledgements, at Page 17 Line 2-3 of the revised manuscript: "John Marsham was funded by the NERC project SWAMMA (NE/L005352/1)."

We add the following at Page 16 Line 27-30 of the revised manuscript: "Since the the methods used by the Met Office in the determination of CTH are also applied in other forms by other satellite applications facilities globally, the relevance of the results presented here are not limited to Met Office products, but may also have implications for other cloud-retrieval algorithms which employ similar techniques."